# Off-Policy Evaluation for Large Action Spaces via Policy Convolution

## ABSTRACT

Developing accurate off-policy estimators is crucial for both evaluating and optimizing for new policies. The main challenge behind the off-policy estimators is the distribution shift between the logging policy which generates the data and the target policy that we aim to evaluate. Typically, techniques for correcting distribution shifts involve some form of importance sampling. This approach results in unbiased value estimation but often comes with the trade-off of high variance, even in the simpler case of one-step contextual bandits. Furthermore, importance sampling relies on the common support assumption, which becomes impractical when the action space is large. To address these challenges, we introduce the Policy Convolution (PC) estimator. This method leverages latent structure within actions—made available through action embeddings—to strategically convolve the logging and target policies. This convolution introduces a unique bias-variance trade-off, which can be controlled by adjusting the *amount of convolution*. Our experiments on synthetic and real-world benchmark datasets demonstrate remarkable mean squared error (MSE) improvements when using PC, especially when either the action space or policy mismatch becomes large, with gains of up to $5-6$ orders of magnitude over existing estimators.

### ACM Reference Format:
Anonymous Author(s). 2018. Off-Policy Evaluation for Large Action Spaces via Policy Convolution. In *Proceedings of Make sure to enter the correct conference title from your rights confirmation emai (Conference acronym 'XX).* ACM, New York, NY, USA, 35 pages. https://doi.org/XXXXXXX.XXXXXXX

## 1 INTRODUCTION

Off-policy estimation (OPE) is a fundamental problem in reinforcement learning and decision making under uncertainty. It involves estimating the expected value of a target policy, given access to only an offline dataset from a different policy, often referred as the logging policy (see [45] for a comprehensive survey). This decoupling between data collection and policy evaluation is crucial in many real-world applications, as it allows for the assessment of new policies using historical data without having to deploy them in the environment, which can be costly or risky. In this paper, we focus on OPE for the one-step contextual bandit setting, *i.e.*, we perform decision making with only an observed context that is

assumed to be independently sampled (*e.g.*, a user coming to a website), and do not consider any recurrent dependencies in the context transitions as is the case in reinforcement learning. A variety of practical applications naturally fall into the off-policy contextual bandit framework, *e.g.*, recommender systems [32, 33], healthcare [22, 23], robotics [25], *etc.*

OPE, in its most general setting, can be a very challenging problem due to its inherently counterfactual nature, as we observe the true reward only for those actions taken by the logging policy, while we aim to evaluate a target policy which might have an arbitrarily different context-action visitation distribution. For example, consider a situation where the logging policy in a movie recommendation service recommends a particular type of action movies 99% of the time, but a target policy—whose value we're trying to estimate—recommends a different type of action movies 99% of the times. Despite both policies recommending movies of the same genre, this distribution-shift, in the worst case, can lead to irrecoverable bias in our estimates [28], making it difficult to accurately evaluate a target policy or learn a better one, *e.g.*, using counterfactual risk minimization [12, 40].

Typical off-policy estimators utilize Importance Sampling (IS) to correct for the policy mismatch between the target and logging policies [4, 10, 14, 20, 29, 37, 39, 41, 44, 48], leading to unbiased value estimation, at the cost of high variance. The variance problem caused by IS is exacerbated if the target and logging policies exhibit significant divergence, and even more so if the action space is large. Notably, large action spaces frequently occur in practical OPE scenarios, *e.g.*, recommender systems which can have millions of items (actions) [27, 49], extreme classification [18, 21, 34], discretized continuous action-spaces [43], *etc.*

To address the aforementioned limitations of IS, we propose the Policy Convolution (PC) estimator. PC strategically convolves the logging and target policies by exploiting the inherent latent-structure amongst actions—available through action-embeddings—to make importance sampling operate in a more favorable bias-variance trade-off region. Such structure can occur naturally in different forms like action meta-data (text, images, *etc.*), action hierarchies, categories, *etc.* Or they can be estimated using domain-specific representation learning techniques [2]. Notably, the utilization of additional action-structure has also been studied in the online multi-armed bandit literature (Lipschitz bandits) [15, 35, 36], albeit in the context of regret minimization.

To better understand the practical effectiveness of PC, we compare its performance with various off-policy estimators on synthetic and real-world benchmark datasets, simulating a variety of off-policy scenarios. Our results demonstrate that PC can skillfully balance the bias-variance trade-off posited by policy convolutions, leading up to $5-6$ orders of magnitude better off-policy evaluation (in terms of MSE), particularly when the action space is large or

the policy mismatch is high. To summarize, *in this paper*, we make the following contributions:

- Introduce the Policy Convolution (PC) off-policy estimator that posits a novel bias-variance trade-off controlled by the *amount of convolution* on the logging and target policies.
- Experiment with four different convolution functions and four different backbone estimators for PC, where each combination is accompanied with its unique set of inductive biases, thereby leading to distinct performance comparisons.
- Conduct empirical analyses on synthetic and real-world benchmark datasets that demonstrate the superiority of PC over a variety of off-policy estimators, especially when the action space or policy mismatch becomes large.

## 2 PRELIMINARIES

### 2.1 OPE in Contextual Bandits

We study OPE in the standard stochastic contextual bandits setting with a context space $\mathcal{X}$ and a finite action space $\mathcal{A}$. In each round $i$, the agent observes a context $x_i \in \mathcal{X}$, takes an action $a_i \in \mathcal{A}$, and observes a reward $r_i \in [0, 1]$. The context $x_i$ is drawn from some unknown distribution $p(x)$. The action $a_i$ follows some policy $\pi(\cdot|x_i)$, and the reward $r_i$ is draw from an unknown distribution $p(r|x_i, a_i)$ with expected value $\delta(a, x) \triangleq \mathbb{E}_{r \sim p(r|a,x)}[r]$. The value of a policy is its expected reward

$$V(\pi) \triangleq \mathbb{E}_{x \sim p(x)}\left[\mathbb{E}_{a \sim \pi(\cdot|x)}[\delta(a, x)]\right].$$

In OPE, given a target policy $\pi$, we aim to estimate its value $V(\pi)$ using some bandit feedback data $\mathcal{D} \triangleq \{(x_i, a_i, r_i)\}_{i=1}^n$ collected under a different policy $\mu$. We call $\mu$ the logging policy and assume it is known.

### 2.2 Conventional OPE Estimators

We now briefly discuss a few prominent OPE estimators, which will also be used to instantiate our proposed Policy Convolution (PC) estimator discussed in Section 3.

*2.2.1 Direct Method (DM).* Taking a model-based approach, DM leverages a reward-model to estimate the value of the target policy. Formally, given a suitable $\hat{\delta} : \mathcal{A} \times \mathcal{X} \mapsto \mathbb{R}$, the estimator is defined as follows:

$$\hat{V}_{\text{DM}}(\pi) \triangleq \mathbb{E}_{(x,\cdot,\cdot) \sim \mathcal{D}}\left[\sum_{a \in \mathcal{A}} \pi(a|x) \cdot \hat{\delta}(a, x)\right],$$

where the outer expectation is over the finite set of logged bandit feedback data $\mathcal{D}$. Notably, the variance of $\hat{V}_{\text{DM}}(\cdot)$ is often quite low, since $\hat{\delta}$ is typically bounded. However, it can suffer from a large bias problem due to model misspecification [5].

*2.2.2 Inverse Propensity Scoring (IPS).* IPS [10] estimator uses Monte-Carlo approximation and importance sampling to account for the policy-mismatch between $\pi$ and $\mu$ as follows:

$$\hat{V}_{\text{IPS}}(\pi) \triangleq \mathbb{E}_{(x,a,r) \sim \mathcal{D}}\left[\frac{\pi(a|x)}{\mu(a|x)} \cdot r\right].$$

The IPS estimator is unbiased under the following two assumptions which we too assume throughout the paper unless otherwise specified:

ASSUMPTION 2.1. **(Unconfoundedness)** *The action selection procedure is independent of all potential outcomes given the context, i.e.,* $\forall x \in \mathcal{X}, \pi \in \Pi, a \sim \pi(\cdot|x); \{\delta(a', x)\}_{a' \in \mathcal{A}} \perp a \mid x.$

ASSUMPTION 2.2. **(Common Support)** *The target policy $\pi$ shares common support with the logging policy $\mu$, if $\forall x \in \mathcal{X}, a \in \mathcal{A}; \pi(a|x) > 0 \implies \mu(a|x) > 0.$*

However, IPS estimator can suffer from a large variance problem, since the importance weights $\pi(a|x)/\mu(a|x)$ can be unbounded and huge. Several estimators are proposed to reduce the variance of the IPS estimator.

*2.2.3 Self-normalized Inverse Propensity Scoring (SNIPS).* Built on the observation that the expected propensity weight in IPS equals 1 , SNIPS [41] uses the empirical average of the propensity weights as a control variate for IPS as follows:

$$\hat{V}_{\text{SNIPS}}(\pi) \triangleq \mathbb{E}_{(x,a,r) \sim \mathcal{D}}\left[\frac{\pi(a|x)}{\rho \cdot \mu(a|x)} \cdot r\right] \text{ s.t. } \rho \triangleq \mathbb{E}_{(x,a,\cdot) \sim \mathcal{D}}\left[\frac{\pi(a|x)}{\mu(a|x)}\right].$$

SNIPS typically enjoys smaller variance at the cost of a slight added bias in comparison to IPS, especially when the variance of the propensity weight is large[8]. Further, $\hat{V}_{\text{SNIPS}}(\pi)$ is a strongly consistent estimator of $V(\pi)$ by the law of large numbers.

*2.2.4 Doubly Robust (DR).* DR combines the benefits of unbiased estimation in IPS and the low-variance, model-based estimation in DM:

$$\hat{V}_{\text{DR}}(\pi) \triangleq \mathbb{E}_{(x,a,r) \sim \mathcal{D}}\left[\frac{\pi(a|x)}{\mu(a|x)} \cdot (r - \hat{\delta}(a, x)) + \Delta(\pi, x)\right]$$

$$\text{s.t. } \Delta(\pi, x) \triangleq \sum_{a' \in \mathcal{A}} \pi(a'|x) \cdot \hat{\delta}(a', x),$$

where $\hat{\delta}$ is the same reward-model as used in DM (Section 2.2.1). Intuitively, DR uses the reward-model as a baseline, and performs importance sampling only on the error of the given reward-model. DR is unbiased and can be of smaller variance than IPS when the reward-model $\hat{\delta}$ is close to the true reward $\delta$ [4].

*2.2.5 Self-normalized Doubly Robust (SNDR).* Similar to the idea behind SNIPS (Section 2.2.3); SNDR [26, 44] performs the same control variate trick on the DR estimator (Section 2.2.4) as follows:

$$\hat{V}_{\text{SNDR}}(\pi) \triangleq \mathbb{E}_{(x,a,r) \sim \mathcal{D}}\left[\frac{\pi(a|x)}{\rho \cdot \mu(a|x)} \cdot (r - \hat{\delta}(a, x)) + \Delta(\pi, x)\right]$$

$$\text{s.t. } \rho \triangleq \mathbb{E}_{(x,a,\cdot) \sim \mathcal{D}}\left[\frac{\pi(a|x)}{\mu(a|x)}\right]; \Delta(\pi, x) \triangleq \sum_{a \in \mathcal{A}} \pi(a|x) \cdot \hat{\delta}(a, x).$$

Hence, SNDR encapsulates the ideas behind all the aforementioned estimators to conduct strongly consistent, low-variance policy value estimation that might perform well (in terms of MSE) in practice.

While effective to some extent, the importance sampling based estimators mentioned above can still suffer from large variance due to large importance sampling weights, especially when the action space is large. In particular, the variance of these importance sampling based estimators grows roughly linearly *w.r.t.* the maximum propensity weight in $\mathcal{D}$. And the maximum propensity

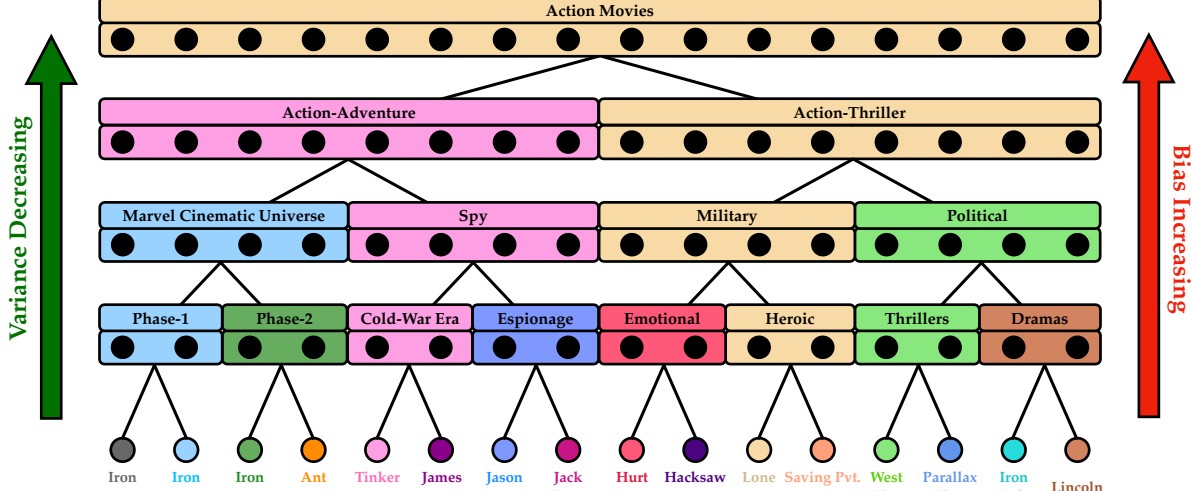

**Figure 1: Intuition for the PC estimator demonstrated using a hierarchical action tree, where similar actions (movies in this example) are recursively agglomerated together. The last level (leaf nodes) represents the complete action space, and the higher levels consist of "meta-actions" that represent a group of individual actions. As we go higher, PC defines the convolved policy for a given action, as the mean probability of all actions inside its corresponding *meta-action*. Hence, we obtain the uniform policy at the topmost-level, and recover the original policy at the last-level.**

weight can grow linearly *w.r.t.* the size of action space $\Omega(|\mathcal{A}|)$ [42], making these estimators undesirable for OPE for large action-space problems. Further, when Assumption 2.2 is violated, the variance of such importance sampling based estimators becomes unbounded, in addition to incurring a bias of $\mathbb{E}_x \left[ \sum_{a \in \mathcal{U}(x)} \pi(a|x)\delta(a|x) \right]$, where $\mathcal{U}(x)$ is the set of actions where $\mu(\cdot|x)$ doesn't put any probability mass on (blind spots) [28].

To address the aforementioned problems of importance sampling based estimators, we introduce policy convolution that makes use of action-similarity information in the next section.

## 3  OPE VIA POLICY CONVOLUTION

In addition to the offline dataset $\mathcal{D}$, we further posit access to some embedding $\mathcal{E} : \mathcal{A} \mapsto \mathbb{R}^d$ of the actions, which maps an action $a$ to a $d$-dimensional embedding space $\mathcal{E}(a) \in \mathbb{R}^d$. Let $\mathcal{E}_{\mathcal{A}} \subset \mathbb{R}^d$ be the subspace spanned by $\mathcal{E}$. Ideally, the embedding should capture action-similarity information, *i.e.*, smaller distance in the embedding space should imply smaller difference in terms of expected reward for each context $x \in \mathcal{X}$. Notably, such action-embeddings are typically readily available in many industrial recommender systems. We are now ready to define our policy convolution estimator, PC-IPS as follows:

$$\hat{V}_{\text{PC-IPS}}(\pi) \triangleq \mathop{\mathbb{E}}_{(x,a,r)\sim\mathcal{D}} \left[ \frac{(\pi(\cdot|x) * f_{\tau_1})(a)}{(\mu(\cdot|x) * f_{\tau_2})(a)} \cdot r \right]$$

$$= \mathop{\mathbb{E}}_{(x,a,r)\sim\mathcal{D}} \left[ \frac{\sum_{a'} \pi(a'|x) \cdot f_{\tau_1}(\mathcal{E}(a), \mathcal{E}(a'))}{\sum_{a'} \mu(a'|x) \cdot f_{\tau_2}(\mathcal{E}(a), \mathcal{E}(a'))} \cdot r \right],$$

where '$*$' represents the convolution operator specified in the action-embedding domain, and $f_\tau : \mathbb{R}^d \times \mathbb{R}^d \mapsto \mathbb{R}$ is an action-action similarity (or pooling) function which has a parameter $\tau$ to

control the level of smoothing. We illustrate the intuition behind policy convolution in Figure 1, where PC "strategically" biases the logging and target policies toward the uniform policy, by leveraging the underlying action structure.

As we will observe further in Section 4.2, such convolutions in-turn lead to a new bias-variance trade-off that is controlled by the level of smoothing ($\tau$). We explore four suitable instantiations of the action-action similarity (or pooling) function, $f_\tau$ for PC:

- **Kernel Smoothing.** Perhaps the most intuitive, we use the idea of multi-variate kernel smoothing [46] in the action-embedding space to derive our similarity function as:

$$f_\tau \left( \mathcal{E}(a), \mathcal{E}(a') \right) \triangleq \mathcal{K} \left( \mathcal{E}(a), \mathcal{E}(a') \right)$$

$$= \frac{1}{\tau^d} \prod_{i=1}^d \mathbf{K} \left( \frac{\mathcal{E}(a)_i, \mathcal{E}(a')_i}{\tau} \right),$$

where, $\mathbf{K}$ is a suitable kernel function (*e.g.*, Gaussian), and $\tau \in \mathbb{R}$ now corresponds to the bandwidth. It is worth noting that such a formulation can also be derived by viewing actions as continuous treatments, as defined by their embeddings [9, 14]. However, in such a formulation, since there cannot exist an inverse mapping from $\mathbb{R}^d \mapsto \mathcal{A}$, having treatments outside $\mathcal{E}_{\mathcal{A}}$ doesn't make sense.

- **Tree Smoothing.** In this setting, we use $\mathcal{E}$ to recursively partition the action-space (see Figure 1 for a depiction) into a tree-like structure of depth $D$, where each depth can be specified by $\mathcal{T}_d \triangleq \{\mathbf{a}_{d,1}, \mathbf{a}_{d,2}, \ldots, \mathbf{a}_{d,k}\}$, where $\mathbf{a}_{d,i}$ is a *meta-action* (set) comprising of singular actions, such that $\mathcal{A} \equiv \bigcup_{i=1}^k \mathbf{a}_{d,i}$, and $\mathbf{a}_{d,i} \cap \mathbf{a}_{d,j} = \Phi$ for all $i \neq j$ pairs. Notably, $\mathcal{T}_D$ (root node) consists of a single meta-action with all actions, and $\mathcal{T}_1 \equiv \mathcal{A}$ (last level) consists of each singular action. The similarity function is then

| | | $a_1$ | $a_2$ | $a_3$ | $a_4$ | $V^*(\cdot)$ | IPS with $|\mathcal{D}| = 10$ | | |
|---|---|---|---|---|---|---|---|---|---|
| | | | | | | | MSE | Bias$^2$ | Var |
| | $\delta(\cdot, x_1)$ | 5 | 10 | 15 | 20 | – | – | – | – |
| $\tau = 1$ | $\pi(\cdot|x_1)$ | 0.0 | 0.2 | 0.2 | 0.6 | 17 | 48.0 | $\approx 0$ | 48.0 |
| | $\mu(\cdot|x_1)$ | 0.2 | 0.2 | 0.4 | 0.2 | 13 | | | |
| $\tau = 2$ | $(\pi * f_\tau)(\cdot|x_1)$ | 0.1 | 0.1 | 0.4 | 0.4 | 15.5 | 13.4 | 4.6 | 8.8 |
| | $(\mu * f_\tau)(\cdot|x_1)$ | 0.2 | 0.2 | 0.3 | 0.3 | 13.5 | | | |
| $\tau = 3$ | $(\pi * f_\tau)(\cdot|x_1)$ | 0.25 | 0.25 | 0.25 | 0.25 | 12.5 | 18.6 | 16 | 2.6 |
| | $(\mu * f_\tau)(\cdot|x_1)$ | 0.25 | 0.25 | 0.25 | 0.25 | 12.5 | | | |

Table 1: A toy example to intuit PC using the Tree similarity function, and constrained to have $\tau_1 = \tau_2$. Similar to Figure 1, in this toy example, the action tree is a 3-level complete binary tree, where the action partitioning is defined as $\{(a_1, a_2, a_3, a_4)\} \rightarrow \{(a_1, a_2), (a_3, a_4)\} \rightarrow \{(a_1), (a_2), (a_3), (a_4)\}$ from top to bottom.

defined as:

$$f_\tau\left(\mathcal{E}(a), \mathcal{E}(a')\right) \triangleq \frac{\mathbb{I}(a' \in \mathbf{a}_\tau(a))}{|\mathbf{a}_\tau(a)|},$$

where, $\mathbb{I}(\cdot)$ represents the indicator function, $\tau$ signifies the depth of the action-tree to use, and $\mathbf{a}_\tau(a)$ represents the meta-action at depth $\tau$ corresponding to the action $a$.

- **Ball Smoothing.** In this setting, we define a binary similarity function based on a fixed-radius ball around the given action, as defined by $\mathcal{E}$ as follows:

$$f_\tau\left(\mathcal{E}(a), \mathcal{E}(a')\right) \triangleq \frac{\mathbb{I}\left(\|\mathcal{E}(a) - \mathcal{E}(a')\|_2^2 < \tau\right)}{\left|\left\{\|\mathcal{E}(a) - \mathcal{E}(a'')\|_2^2 < \tau \mid \forall a'' \in \mathcal{A}\right\}\right|},$$

where, $\tau$ signifies the radius of the ball around $\mathcal{E}(a)$.

- **kNN Smoothing.** In this setting, we define a binary similarity function as the k-nearest neighbors decision function:

$$f_\tau\left(\mathcal{E}(a), \mathcal{E}(a')\right) \triangleq \frac{\mathbb{I}\left(a' \in \text{kNN}(a, \tau)\right)}{\tau},$$

where, $\tau$ signifies the number of nearest neighbors to use, and $\text{kNN}(a, \tau)$ represents the set of $\tau$ nearest neighbors of $\mathcal{E}(a)$ in $\mathcal{E}_{\mathcal{A}}$.

Notably, PC is not limited for the IPS estimator discussed hitherto, and we analogously define PC for other "backbone" estimators, namely, Self-Normalized IPS (SNIPS), Doubly Robust (DR), Self-Normalized DR (SNDR) discussed in Sections 2.2.3 to 2.2.5, and call such estimators PC-SNIPS, PC-DR, and PC-SNDR for convenience.

Witnessing existing solutions for conducting OPE in large action-spaces [24, 30], we note that offCEM [30] and groupIPS [24] are both specific instantiations of our more general Policy Convolution framework. More specifically, groupIPS can be generalized as PC-IPS and offCEM as PC-DR, both using a two-depth tree (i.e., flat clustering) in the tree convolution function, and with an additional constraint of $\tau_1 = \tau_2 = 1$. As we will further note in our experiments (Section 4.2): (1) using the kernel and kNN convolution functions tend to perform better than others; and (2) convolving the logging and target policies *differently* (i.e., $\tau_1 \neq \tau_2$) adds a lot of flexibility to PC, leading to much better estimation than either convolving the two policies equally, or convolving only one out of the two policies.

**Motivating example.** To gain a better intuition of PC, we refer to Figure 1 and construct a four-action, single-context toy example described in Table 1. We conduct OPE using the IPS estimator at various levels of the action-tree, with a sample size of $|\mathcal{D}| = 10$ and repeat the experiment 50k times. The results demonstrate that as we progress to higher levels of the tree (increased pooling), variance decreases, but bias increases. At the leaf level, IPS is unbiased but exhibits high variance. At the top-most level, while variance is the lowest, bias is significantly increased. When $\tau = 2$, we observe the best bias-variance trade-off, leading to the lowest MSE.

## 4 EXPERIMENTS

### 4.1 Setup

We measure PC's empirical effectiveness on two datasets. Firstly, we simulate a synthetic contextual bandit setup, beginning by sampling contexts $x \sim \mathcal{N}(0, I)$. Subsequently, to realize our assumption that there indeed exists a latent structure amongst individual actions, we randomly assign each action to one of 32 latent *topics*, denoted by $\tilde{a}$. We also assign each latent topic a corresponding mean $\{\mu_i \sim \mathcal{N}(0, I)\}_{i=1}^{32}$ and covariance $\{\sigma_i \sim \mathcal{N}(0, I)\}_{i=1}^{32}$. To realize the assigned structure in the action-space, we sample each action's embedding from its correspondingly assigned topic, i.e., $\mathcal{E}(a) \sim \mathcal{N}(\mu_{\tilde{a}}, \sigma_{\tilde{a}})$. We then model the reward function $\delta(x, a)$ as a noisy and non-linear function of the underlying context- and action-embedding: $\delta(x, a) \triangleq \Phi(x \| \mathcal{E}(a) \| \epsilon)$, where "$\|$" represents concatenation, $\epsilon \sim \mathcal{N}(0, I)$ is white-noise, and $\Phi$ is a randomly initialized, two-layer neural network. Such a formulation realizes two crucial assumptions: (a) semantically closer actions are nearby according to $\mathcal{E}$, and (b) $\mathcal{E}$ shares a causal connection with the downstream reward function. Finally, we define the logging policy $\mu(\cdot|x)$ as a temperature-activated softmax distribution on the ground-truth reward distribution $\delta(\cdot, x)$, and the target policy as the $\epsilon$−greedy policy:

$$\mu(a|x) \triangleq \frac{\exp(\beta \cdot q(x, a))}{\sum_{a'} \exp(\beta \cdot q(x, a'))}$$

$$\pi(a|x) \triangleq (1 - \epsilon) \cdot \mathbb{I}\left(\delta(a, x) = \sup_{a' \in \mathcal{A}} \{\delta(a', x)\}\right) + \frac{\epsilon}{|\mathcal{A}|} \quad (1)$$

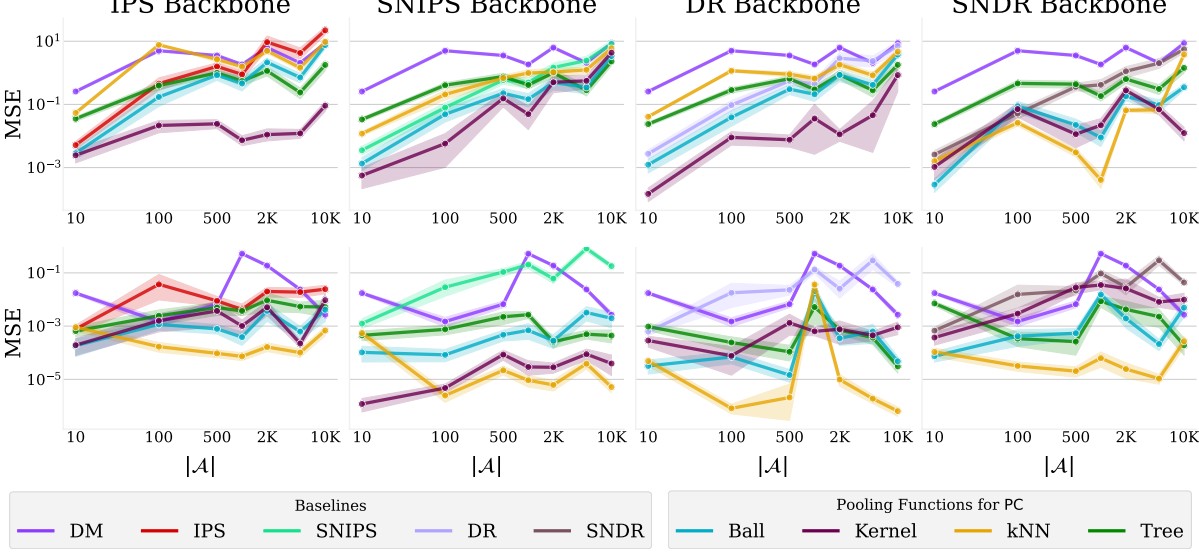

**Figure 2: Change in MSE while estimating $V(\pi_{\text{good}})$ with varying number of actions (log-log scale) for the synthetic dataset, using data logged by (top) $\mu_{\text{uniform}}$, or (bottom) $\mu_{\text{good}}$. Results for $V(\pi_{\text{bad}})$ can be found in Appendix B, Figures 7 and 8.**

Furthermore, to test the practicality of PC on a real-world, large-scale data, we also synthesize a bandit-variation of the Movielens-100k dataset [7] which consists of numerous (user,item,rating) tuples. Taking inspiration from previous recommender system → bandit feedback conversion setups [31], we define a positive reward if the provided rating ≥ 4, or else zero. We then define contexts and action-embeddings as the user- and item-factors attained by performing SVD on the binary user-item rating matrix, respectively. Furthermore, to simulate continuous instead of binary reward, for missing entries, we define the reward as the dot product of the corresponding user- and item-factors, estimated using SVD before. While we define the target policy similarly as in Equation (1), in the hope of following a realistic, two-stage recommender system setup [19], we define the logging policy $\mu(\cdot|x)$ using the following procedure: (1) shortlist a set of 100 best actions defined by $\delta(\cdot, x)$, and 400 actions at random, (2) sample a logit from $U(0, 1)$ for each positive action, and from $U(0, 0.8)$ for the random actions, (3) take a temperature softmax as in Equation (1) only on the sampled logits, and (4) perform $\epsilon$−greedy on the obtained action probabilities to satisfy Assumption 2.2.

As per our setup, $V^*(\mu) \propto \beta$ and $V^*(\pi) \propto \epsilon^{-1}$ for both the datasets. To avoid clutter, we define $\mu_{\text{uniform}}$ when $\beta = 0$, and $\mu_{\text{good}}$ when $\beta = 3$. Similarly, we define $\pi_{\text{bad}}$ when $\epsilon = 0.8$, and $\pi_{\text{good}}$ when $\epsilon = 0.05$. Unless specifically mentioned, we use the default values of the remaining hyper-parameters in the bandit data generation procedure, as listed in Appendix A.

For evaluating the performance of various estimators, we compute the Mean Squared Error (MSE) between the true and predicted value of the target policy. We reserve a large test-set just to compute the true value of the target policy. We also estimate the squared bias and variance of our predicted estimates by repeating each experiment for 50 random seeds, and also compute the 95% confidence

interval for visualization purposes. Note that the bias, variance, and MSE of any estimator are naturally linked to each other by the following decomposition: $\text{MSE}(\cdot) = \text{Bias}(\cdot)^2 + \text{Var}(\cdot)$.

For the PC estimator, we chose the optimal convolution values (i.e., $\tau_1$ and $\tau_2$) using the MSE obtained on a validation set. Notably, while PC for any given backbone estimator strictly contains the naïve backbone (i.e., when $\tau_1 = \tau_2 = 0$); we only report results for PC with a non-negative amount of pooling.

## 4.2 Results

**(Figure 2) How does PC perform with a varying number of actions?** Observing the effect of increasing the number of actions ($|\mathcal{A}|$) on different estimators' performance while keeping the size of the available bandit feedback ($|\mathcal{D}|$) constant, we find that the MSE of all IS-based estimators deteriorates, in accordance with the $\Omega(|\mathcal{A}|)$ growth of their variance. We further note that utilizing $\mu_{\text{good}}$ rather than $\mu_{\text{uniform}}$ as the logging policy, results in an almost 2-orders of magnitude reduction of MSE across all estimators. We hypothesize this finding to be an artifact of the increased overlap between $\mu_{\text{good}}$ and $\pi_{\text{good}}$. Finally, a general trend that holds across various convolution strategies is that the improvement in MSE achieved by PC w.r.t. its respective backbone, significantly increases with increasing $|\mathcal{A}|$. Notably, in the extreme scenario of 10k actions, PC-DR outperforms DR by up to 5-orders of magnitude in terms of MSE. While we note that no single backbone estimator or pooling strategy is optimal in every scenario for PC, PC-SNDR and kernel or kNN convolution strategies generally exhibit better performance than other combinations.

**(Figure 3) How does PC perform with varying amount of policy-mismatch?** We analyze the impact of increasing the policy mismatch between the target and logging policies on off-policy estimation performance, specifically by tuning the $\beta$ parameter in

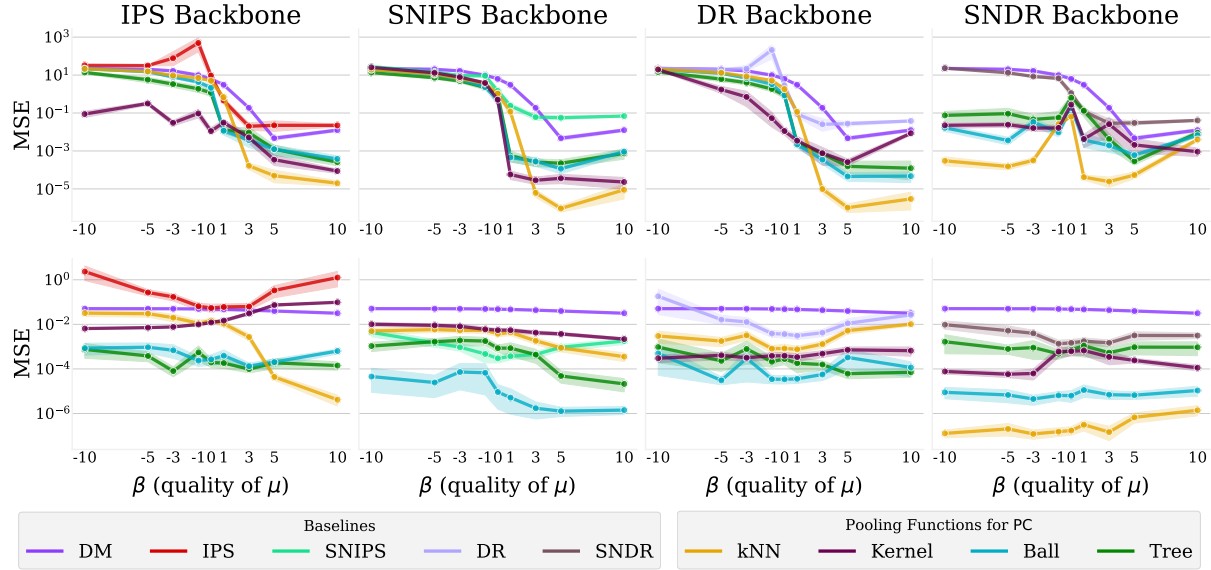

**Figure 3: Change in MSE while estimating $V(\pi_{\text{good}})$ with varying policy-mismatch (log scale) for (top) synthetic, and (bottom) movielens dataset. The policy-mismatch is higher when $\beta$ is lower. Results for estimating $V(\pi_{\text{bad}})$, and the observed bias-variance trade-off can be found in Appendix B, Figures 9 to 12.**

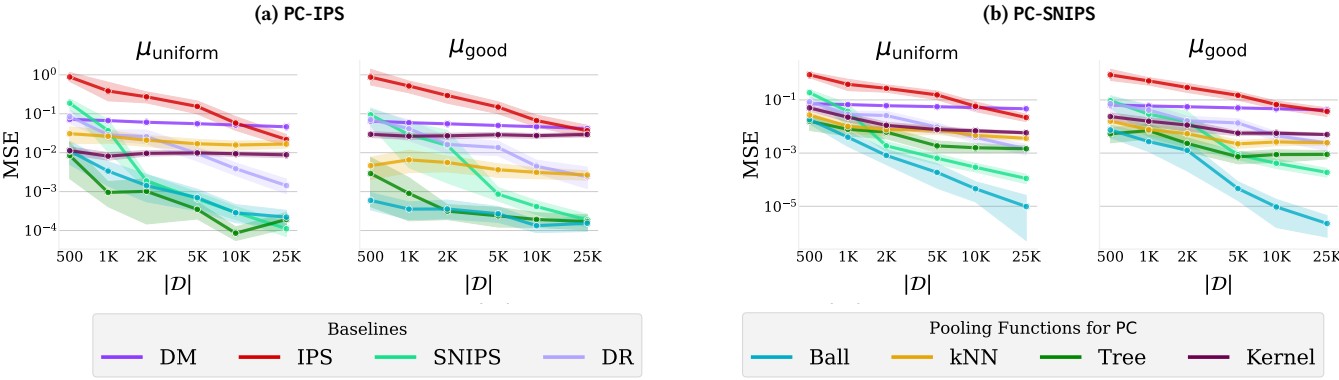

**Figure 4: Change in MSE while estimating $V(\pi_{\text{good}})$ with varying amounts of bandit feedback (log-log scale) for the movielens dataset. Results for PC-DR, PC-SNDR, estimating $V(\pi_{\text{bad}})$, and the synthetic dataset can be found in Appendix B, Figures 17 to 20.**

Equation (1) that controls the quality of the logging policy, keeping the target policy fixed. We observe that OPE becomes hardest on both ends of the spectrum, *i.e.*, when the logging and target policies have large divergence. The difficulties of OPE in such low-overlap scenarios have been well documented in the literature [1, 13, 28], and we observe that PC—through the use of latent structure amongst actions—is particularly helpful in such conditions. An analysis of the exact bias-variance trade-off provided in Appendix B, Figure 10 reveals that PC is able to effectively (1) reduce the variance when policy-mismatch is low, *i.e.*, $|\beta| \approx 0$, and (2) counteract the bias introduced by IS when Assumption 2.2 is violated, *i.e.*, $|\beta|$ is large. Both of these improvements result in significantly better MSE for PC across the entire policy-mismatch spectrum, as depicted in Figure 3.

**(Figure 4) How does PC perform with a varying amount of bandit feedback?** Keeping all other factors fixed, we investigate the impact of increasing the size of the logged bandit feedback ($|\mathcal{D}|$) on the MSE of various off-policy estimators. Like other baseline estimators, we observe that PC exhibits consistency and is effectively able to balance the bias-variance trade-off. Specifically, PC is most advantageous in the low-data regime, due to its variance reduction properties. However, as $|\mathcal{D}|$ continues to increase, PC converges to its respective backbone estimator, *i.e.*, when $\tau_1 = \tau_2 = 0$ represents the optimal bias-variance trade-off point. This pattern of a decreasing amount of optimal pooling ($\tau_1, \tau_2$) with increasing $|\mathcal{D}|$ is anticipated, as the variance of IS-based estimators naturally decreases with growing $|\mathcal{D}|$, and any reduction in variance at the cost of increased bias negatively impacts the overall MSE. We take

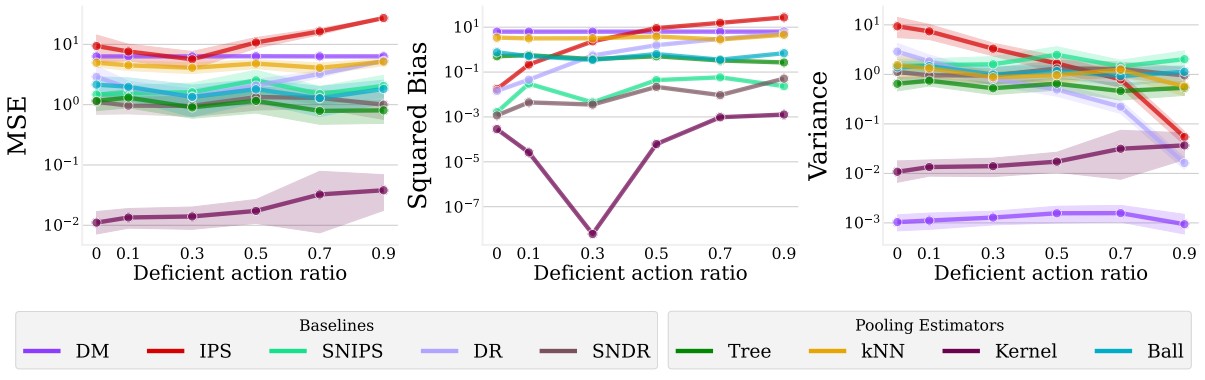

**Figure 5: Change in MSE, Squared Bias, and Variance for PC-IPS and other baseline estimators while estimating $V(\pi_{\text{good}})$ with varying support (log-log scale) for the synthetic dataset (with 2000 actions), using data logged by $\mu_{\text{uniform}}$. Results for other backbones in PC, and estimating $V(\pi_{\text{bad}})$ can be found in Appendix B, Figures 21 and 22.**

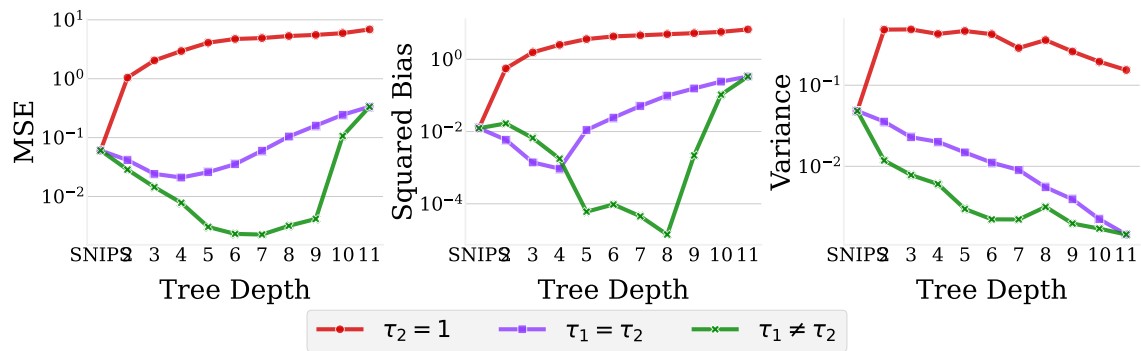

**Figure 6: Visualizing the bias-variance trade-off for PC-SNIPS (Tree pooling) with varying amount of pooling, while estimating $V(\pi_{\text{good}})$ and using $\mu_{\text{good}}$ for logging on the synthetic dataset (with 2000 actions). When $\tau_1 \neq \tau_2$, we plot results for $\tau_1$ on the plot. The naïve SNIPS estimator is the left-most point, i.e., when there's no pooling. Results for other backbones, pooling methods, and the movielens dataset can be found in Appendix B, Figures 27 and 28.**

further note of this observation for even more kinds of logging policies and backbone estimators in Appendix B, Figures 29 to 31.

**(Figure 5) How does PC perform with a varying amount of deficient support?** To understand the effect of varying amount of support (or overlap) between the logging and target policies on various estimators, we simulate such a scenario by explicitly forcing the logging policy to only have support (i.e., non-zero probability) over a smaller, random set of actions, and have zero probability for all other actions. Varying this deficient action ratio, we firstly observe an expected increase in the MSE and squared bias of baseline estimators like IPS, SNIPS, DR, etc. due to the violation of Assumption 2.2, which has been shown to to add irrecoverable bias in importance sampling based estimators [28]. On the other hand, even with an increasing number of deficient actions, the MSE for PC tends to stay relatively constant, with kernel-based convolutions being the best approach. This goes to show that PC is able to accurately leverage action-embeddings as a guide for appropriately filling-in the blind spots while performing OPE.

**(Figure 6) How does the amount of convolution affect the bias-variance trade-off?** We examine the influence that the amount of convolution in PC has on the bias-variance trade-off for three variations of PC with the tree pooling function: (1) only the target policy is convolved, i.e., $\tau_2 = 1$ which is equivalent to the similarity estimator [6]; (2) both the logging and target policies are convolved equally, i.e., $\tau_1 = \tau_2$ which is equivalent to the of-fCEM [30] and groupIPS [24] estimators; and (3) both the logging and target policies are convolved and $\tau_1, \tau_2$ can be different. From Figure 6, we observe that the amount of convolution results in a bias-variance trade-off, where larger pooling leads to decreased variance, but increased bias. It is worth mentioning that the initial decrease in bias with convolution is due to the use of $\mu_{\text{good}}$ for logging, which partially violates Assumption 2.2. This results in a biased SNIPS estimate, which the PC estimator is able to effectively recover. Further, we note that solely convolving the target policy (i.e., the similarity estimator) does not necessarily result in a suitable bias-variance trade-off, with the other two convolution strategies being significantly better, and having $\tau_1 \neq \tau_2$ consistently being the best approach.

## 5 RELATED WORK

**Off-policy evaluation.** A wide body of literature in operations research, causal inference, and reinforcement learning studies the problem of off-policy evaluation. Prominent off-policy estimators can be grouped into the following three categories: **(1) Model-based:** dubbed as the direct method (DM), the key idea in DM is to use a parametric reward model to extrapolate the reward for unobserved (context, action) pairs [4]. These methods typically have a very low variance, at the cost of uncontrollable bias. **(2) Importance sampling based:** these estimators use the propensity ratio between the target and logging policies to account for the distribution mismatch [10]. Such estimators, though unbiased, suffer from large variance, with typical remedies being propensity clipping [11, 39] or self-normalization [41]. **(3) Doubly robust:** these estimators use importance sampling only on the error of a reward model leading to unbiased estimation with a much lower variance [3–5, 26, 37, 44].

**Off-policy evaluation for large action spaces.** Most relevant to this paper, two kinds of major problems occur when attempting to perform OPE in large action spaces. Firstly, the variance of any importance sampling method grows linearly *w.r.t.* the size of the action-space [42], and the common support assumption tends to become impractical [28] leading to irrecoverable bias in estimation. Recent work [6, 24, 29, 30] attempts to use some notion of latent structure in the action-space to address both of the aforementioned limitations. The MIPS estimator [29] builds on the randomness in available action-embeddings to improve OPE. However, in a setting where only a 1:1 action-to-embedding mapping is available (as in this paper), MIPS reduces to vanilla IPS. Further, as we discussed in Section 3, offCEM [30], similarity estimator [6], and groupIPS [24] are all specific instantiations of our proposed PC estimator.

**Off-policy evaluation for continuous action spaces.** Another line of work builds off-policy estimators when the action-space is continuous, *e.g.*, the dosage of a treatment. If we discretize the action-space into a fixed number of bins as per some resolution, the action-space becomes too large for typical off-policy estimators to work well [43]. To this end, typical approaches extend the discrete rejection sampling idea into a smooth rejection operation using standard kernel functions [14, 16, 47]. Notably, in PC, we marry the two disparate fields of (1) OPE with continuous action spaces, and (2) OPE with large action spaces through a simple connection: viewing action embeddings as continuous treatments themselves. In this way, we can reliably perform off-policy evaluation in the continuous, low-dimensional embedding space rather than the discrete, large action space.

## 6 CONCLUSION & FUTURE WORK

In this paper, we proposed the Policy Convolution (PC) estimator which leverages latent action structure specified via action embeddings to perform better off-policy evaluation. More specifically, PC convolves both the target and logging policies according to an action-action convolution function, which posits a new kind of bias-variance tradeoff controlled by the amount of convolution.

Conducting empirical evaluation over a diverse set of off-policy estimation scenarios, we observe that the PC estimator is up to

5 orders of magnitude better than other baseline estimators in terms of MSE, especially when (1) the action-space is large, (2) the policy mismatch between logging and target policies is high, or (3) the common support assumption for importance sampling is violated. We believe that our findings can expand the potential use of off-policy estimators into new and practical scenarios, and also encourage further exploration into the use of additional structure for efficient OPE.

We also discuss limitations and unexplored directions in this paper that we believe are promising directions for future work. Firstly, having a deeper formal understanding about the statistical properties of PC might help in designing more robust off-policy estimators. Next, even though we experiment with four different action convolution (or pooling) functions, having a better understanding of the downstream effect of various convolution functions on the PC estimator might guide us in designing even better and more principled approaches. Finally, trying to understand and develop principled techniques for automatically selecting the level of convolution to conduct on the target and logging policies is an interesting research direction [17, 38].

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

## A  APPENDIX: HYPER-PARAMETERS

| Hyper-param | Value | Hyper-param | Value | Hyper-param | Value |
|---|---|---|---|---|---|
| $|\mathcal{A}|$ | 2,000 | $|\mathcal{D}|$ | 10,000 | $|$ Test data $|$ | 100,000 |
| $\beta$ | 0.0 | $\epsilon$ | 0.05 | # Seeds | 50 |
| dim(context) | 32 | dim(action-embed) | 16 | dim(noise) | 8 |

## B  APPENDIX: ADDITIONAL RESULTS

Figure 7: Change in MSE, Squared Bias, and Variance while estimating $V(\pi_{\text{bad}})$ with varying number of actions (log-log scale) for the synthetic dataset, using data logged by $\mu_{\text{uniform}}$.

Figure 8: Change in MSE, Squared Bias, and Variance while estimating $V(\pi_{\text{bad}})$ with varying number of actions (log-log scale) for the synthetic dataset, using data logged by $\mu_{\text{good}}$.

**Figure 9: Change in MSE, Squared Bias, and Variance while estimating $V(\pi_{\text{good}})$ with varying policy-mismatch ($\log$ scale) for the synthetic dataset.**

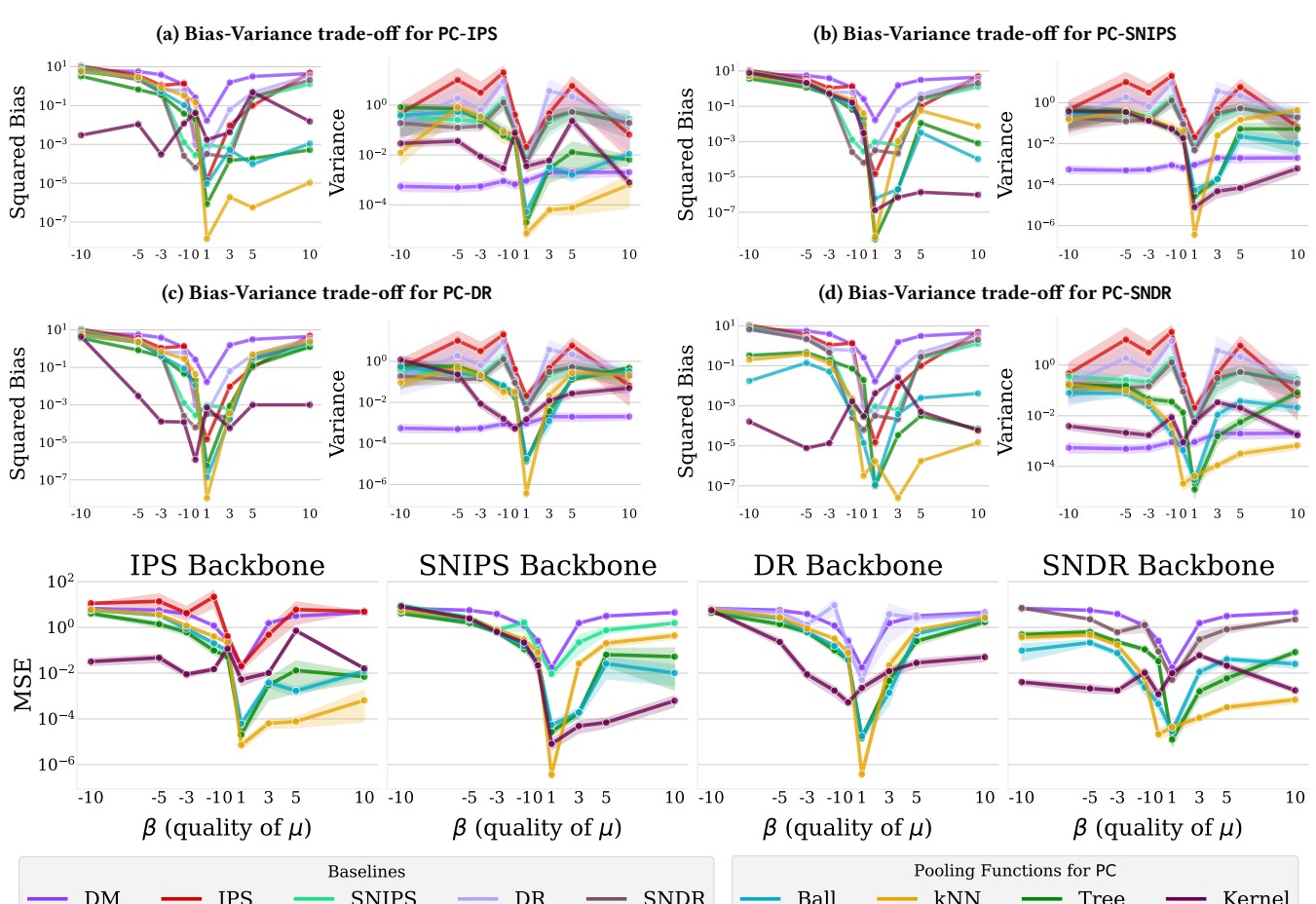

**Figure 10: Change in MSE, Squared Bias, and Variance while estimating $V(\pi_{\text{bad}})$ with varying policy-mismatch ($\log$ scale) for the synthetic dataset.**

**Figure 11: Change in MSE, Squared Bias, and Variance while estimating $V(\pi_{\text{good}})$ with varying policy-mismatch ($\log$ scale) for the movielens dataset.**

Figure 12: Change in MSE, Squared Bias, and Variance while estimating $V(\pi_{\text{bad}})$ with varying policy-mismatch (log scale) for the movielens dataset.

**Figure 13: Change in MSE, Squared Bias, and Variance while estimating various target policies (log scale) for the synthetic dataset, using data logged by $\mu_{\text{uniform}}$.**

Figure 14: Change in MSE, Squared Bias, and Variance while estimating various target policies ($\log$ scale) for the synthetic dataset, using data logged by $\mu_{\text{good}}$.

Figure 15: Change in MSE, Squared Bias, and Variance while estimating various target policies (log scale) for the movielens dataset, using data logged by $\mu_{\mathsf{uniform}}$.

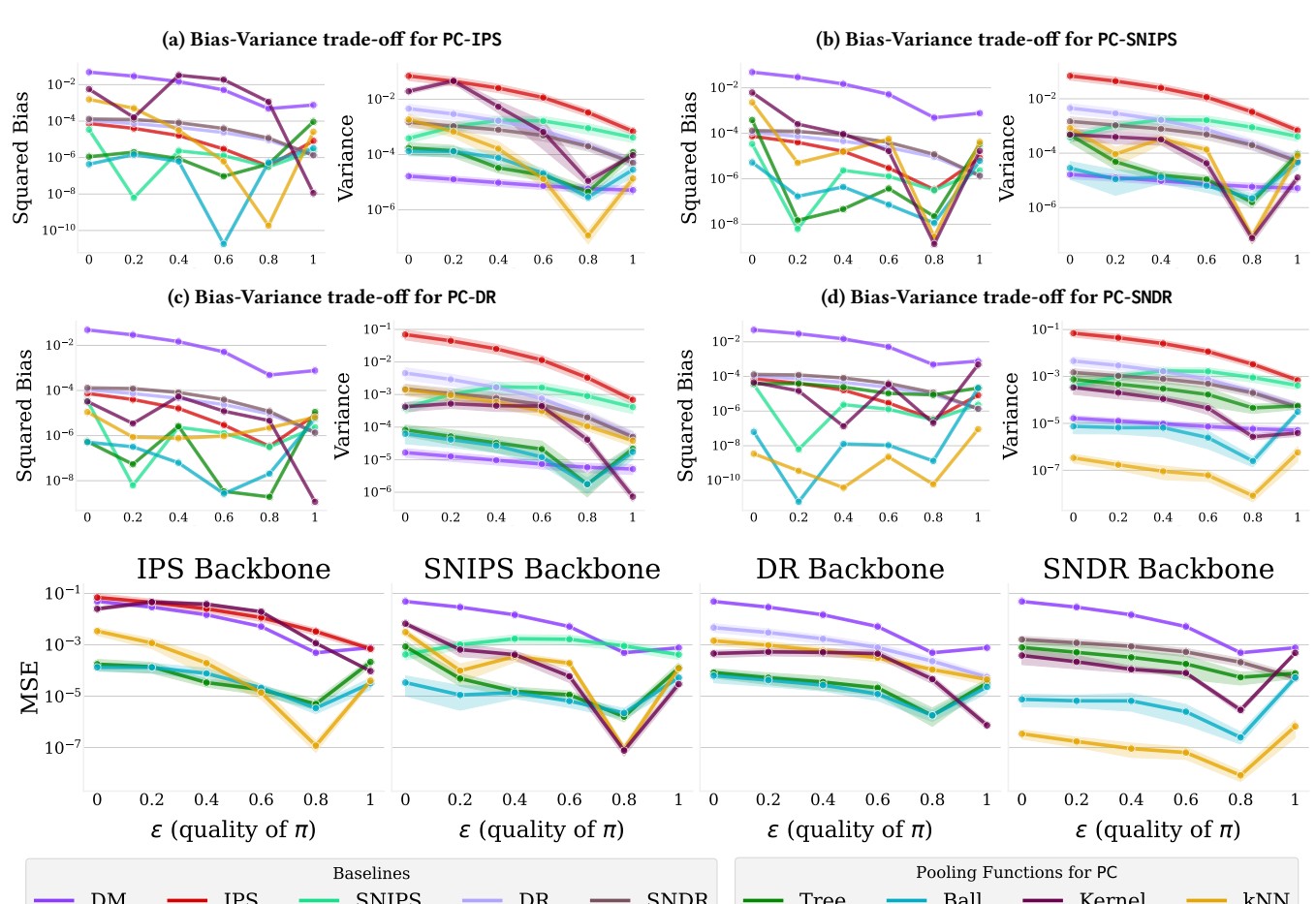

Figure 16: Change in MSE, Squared Bias, and Variance while estimating various target policies (log scale) for the movielens dataset, using data logged by $\mu_{\text{good}}$.

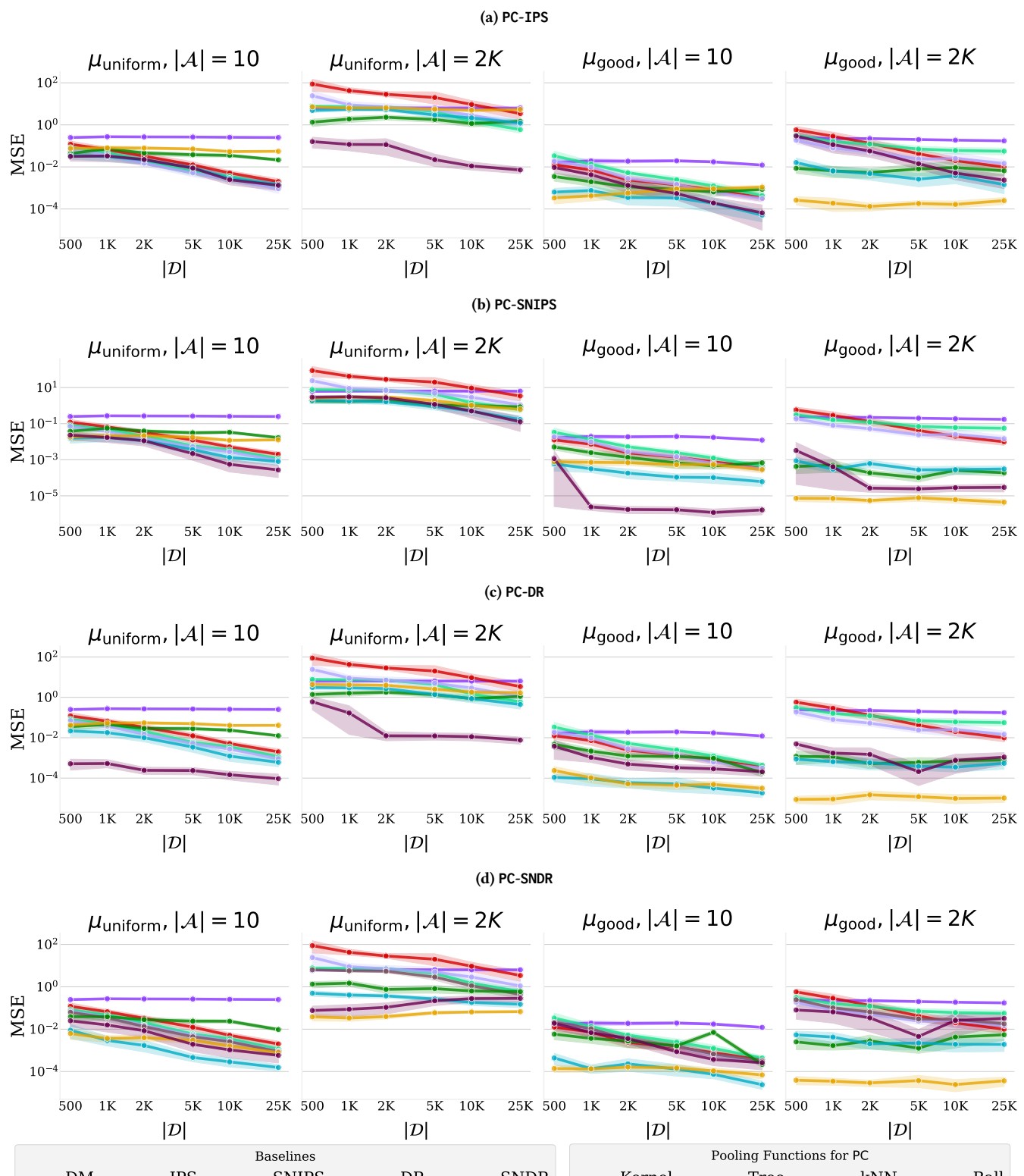

Figure 17: Change in MSE while estimating $V(\pi_{\text{good}})$ with varying amount of bandit feedback (log-log scale) for the synthetic dataset.

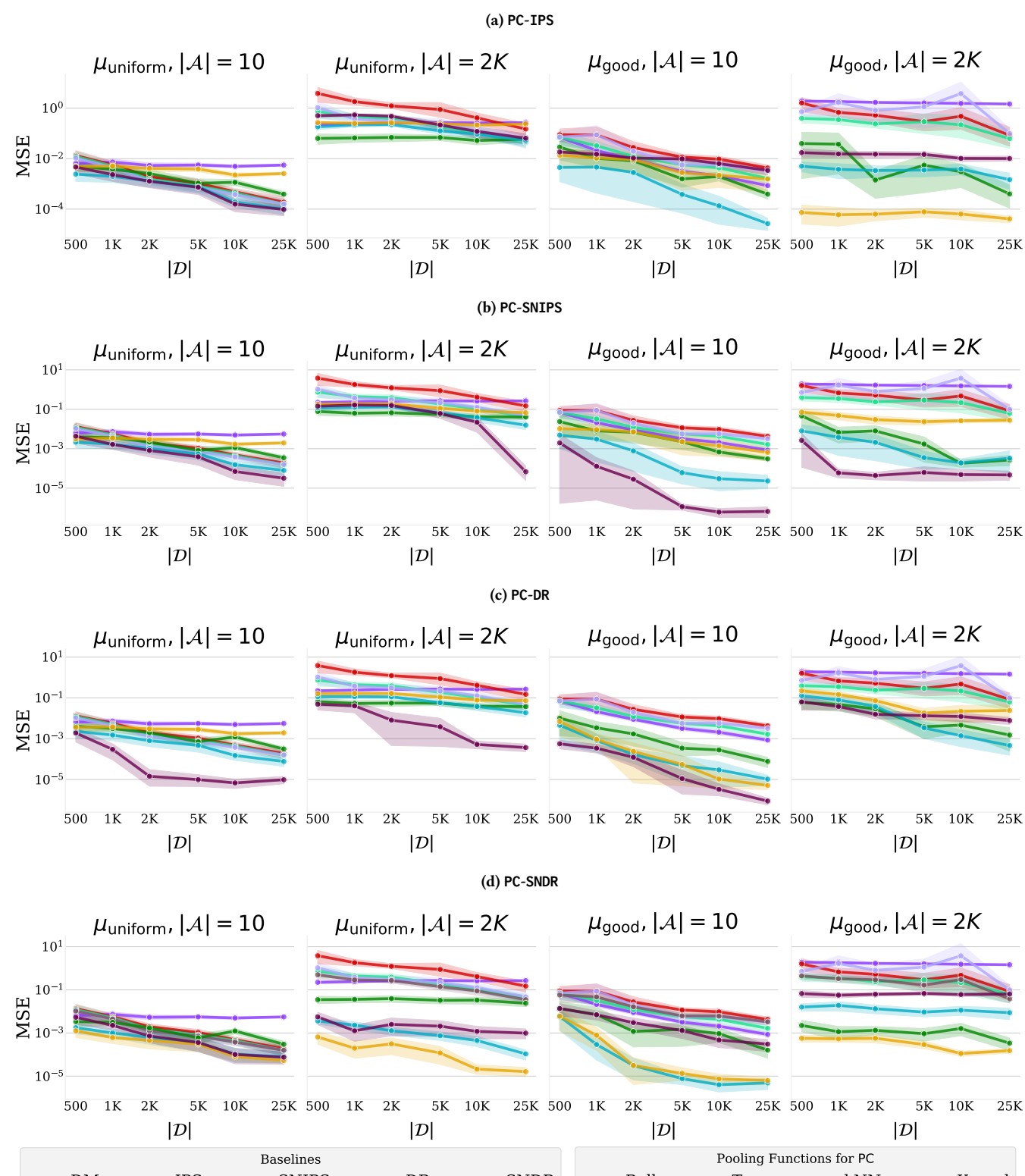

Figure 18: Change in MSE while estimating $V(\pi_{\text{bad}})$ with varying amount of bandit feedback (log-log scale) for the synthetic dataset.

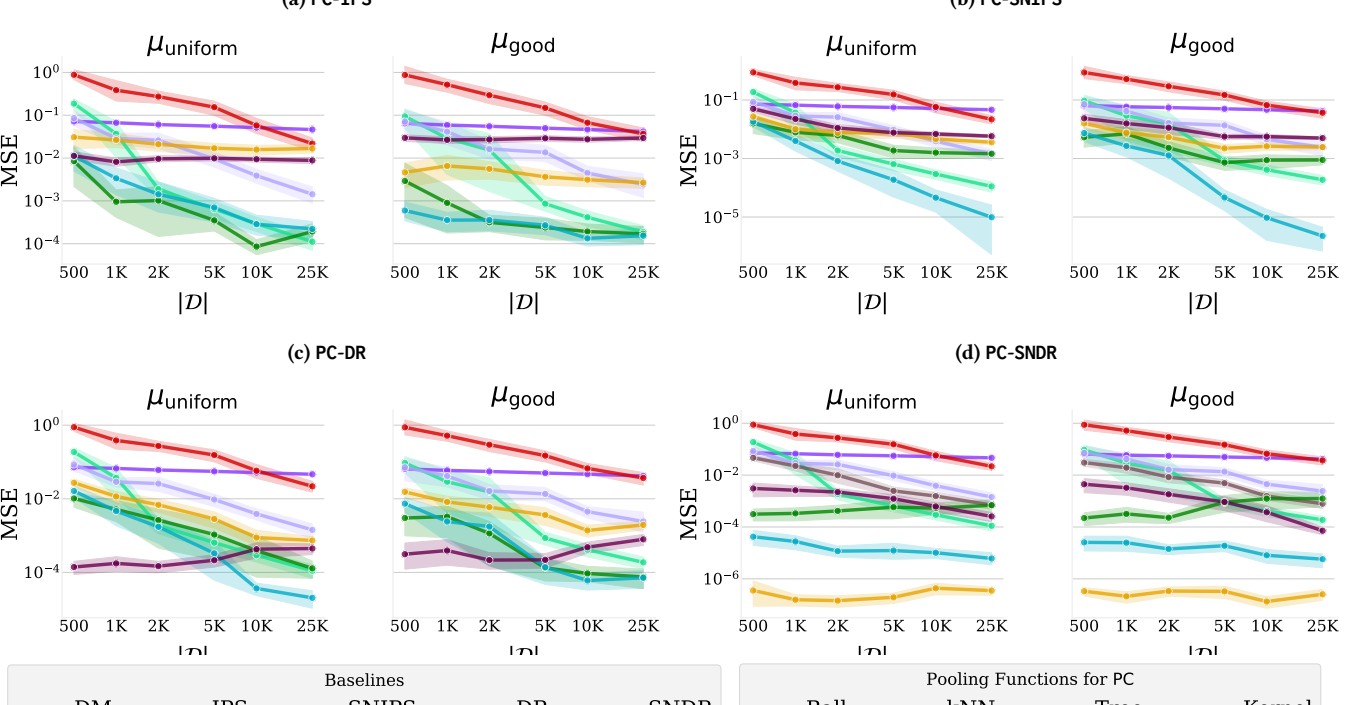

**Figure 19: Change in MSE while estimating $V(\pi_{\text{good}})$ with varying amount of bandit feedback (log-log scale) for the movielens dataset.**

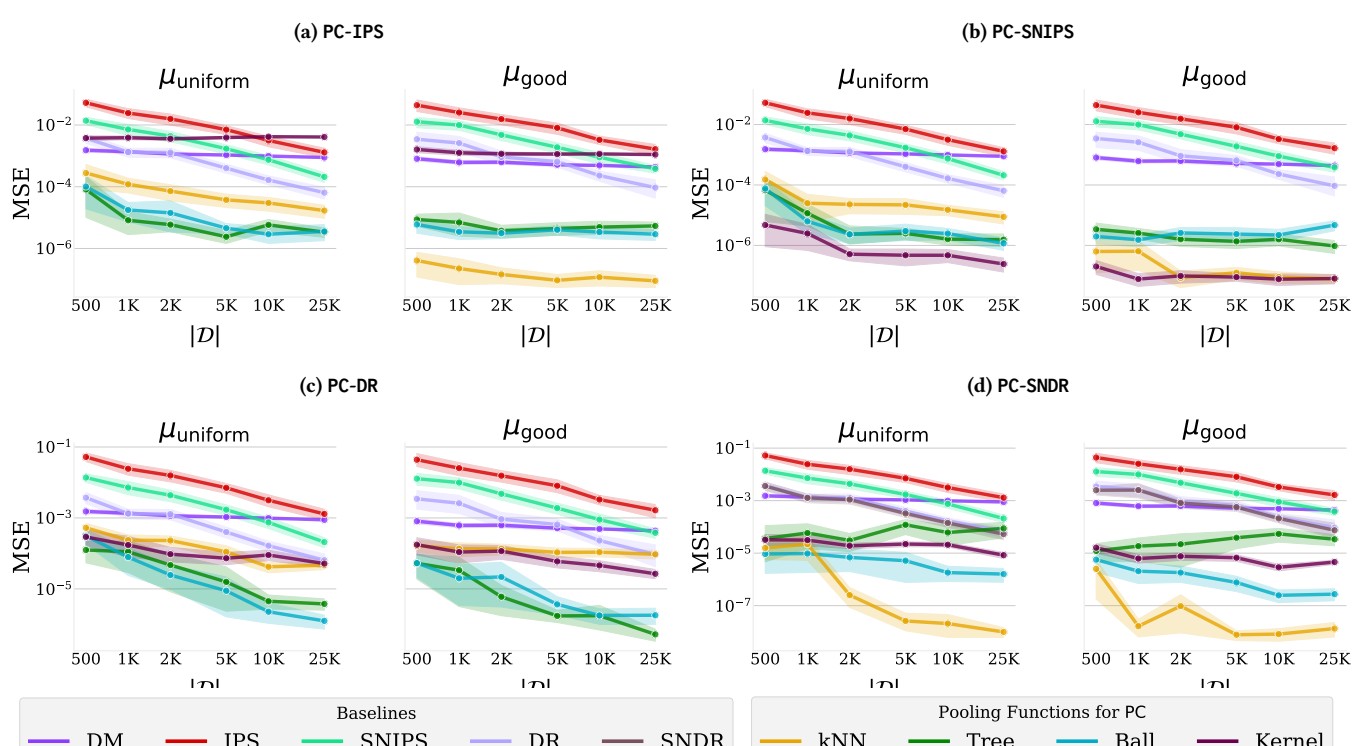

Figure 20: Change in MSE while estimating $V(\pi_{\text{bad}})$ with varying amount of bandit feedback (log-log scale) for the movielens dataset.

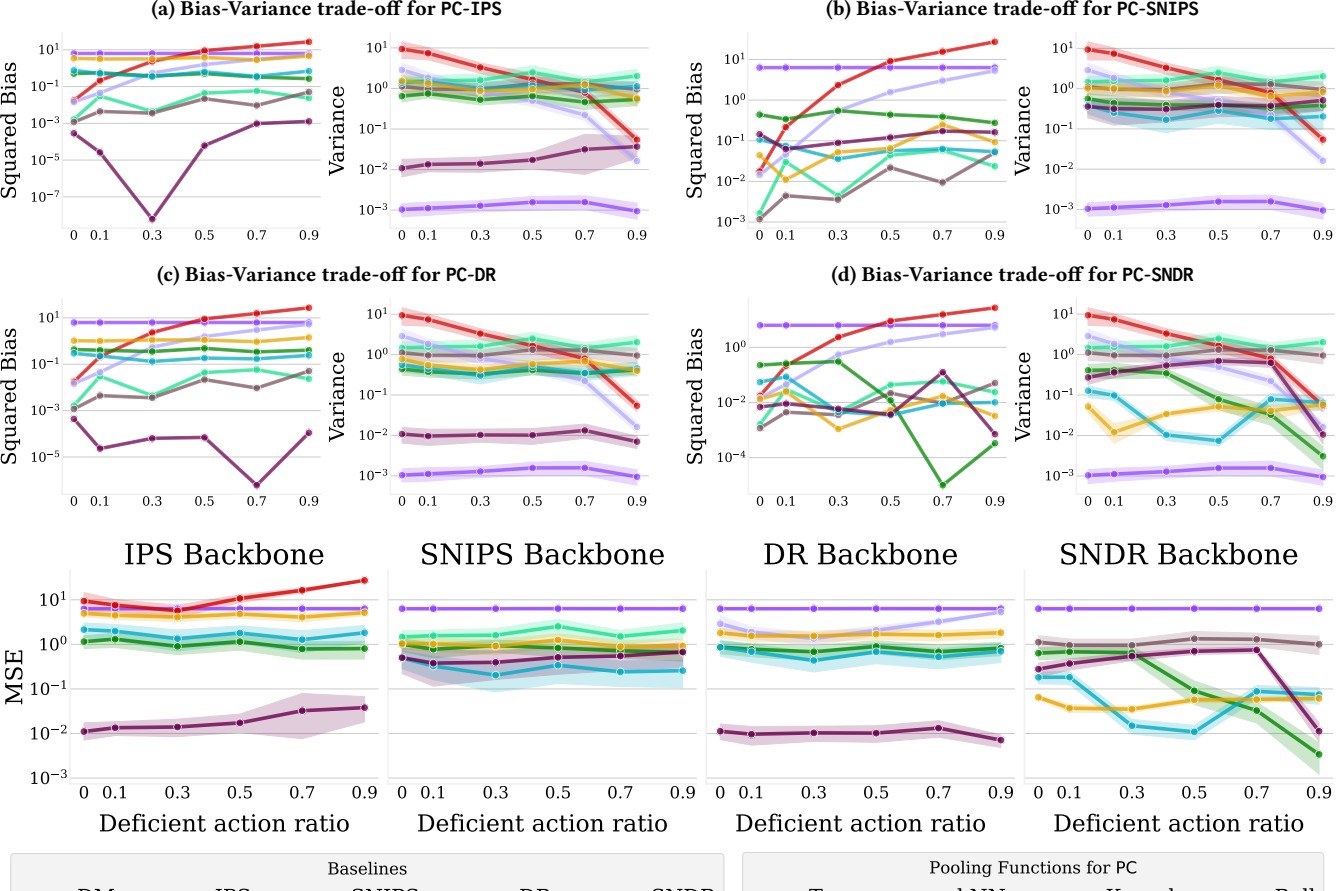

**Figure 21: Change in MSE, Squared Bias, and Variance while estimating $V(\pi_{\text{good}})$ with varying support (log-log scale) for the synthetic dataset (with 2000 actions), using data logged by $\mu_{\text{uniform}}$.**

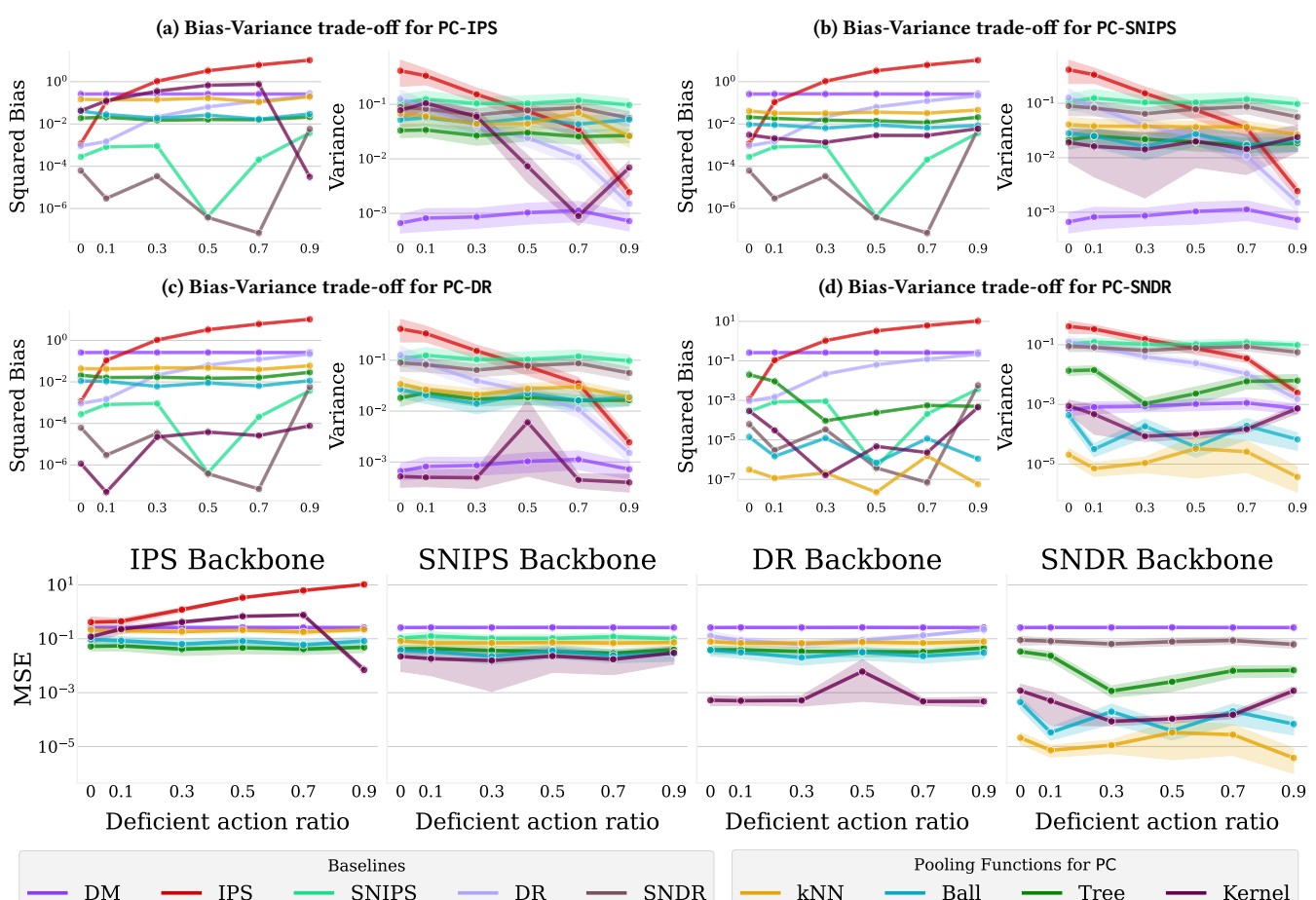

**Figure 22: Change in MSE, Squared Bias, and Variance while estimating** $V(\pi_{\text{bad}})$ **with varying support (**log-log **scale) for the synthetic dataset (with** 2000 **actions), using data logged by** $\mu_{\text{uniform}}$.

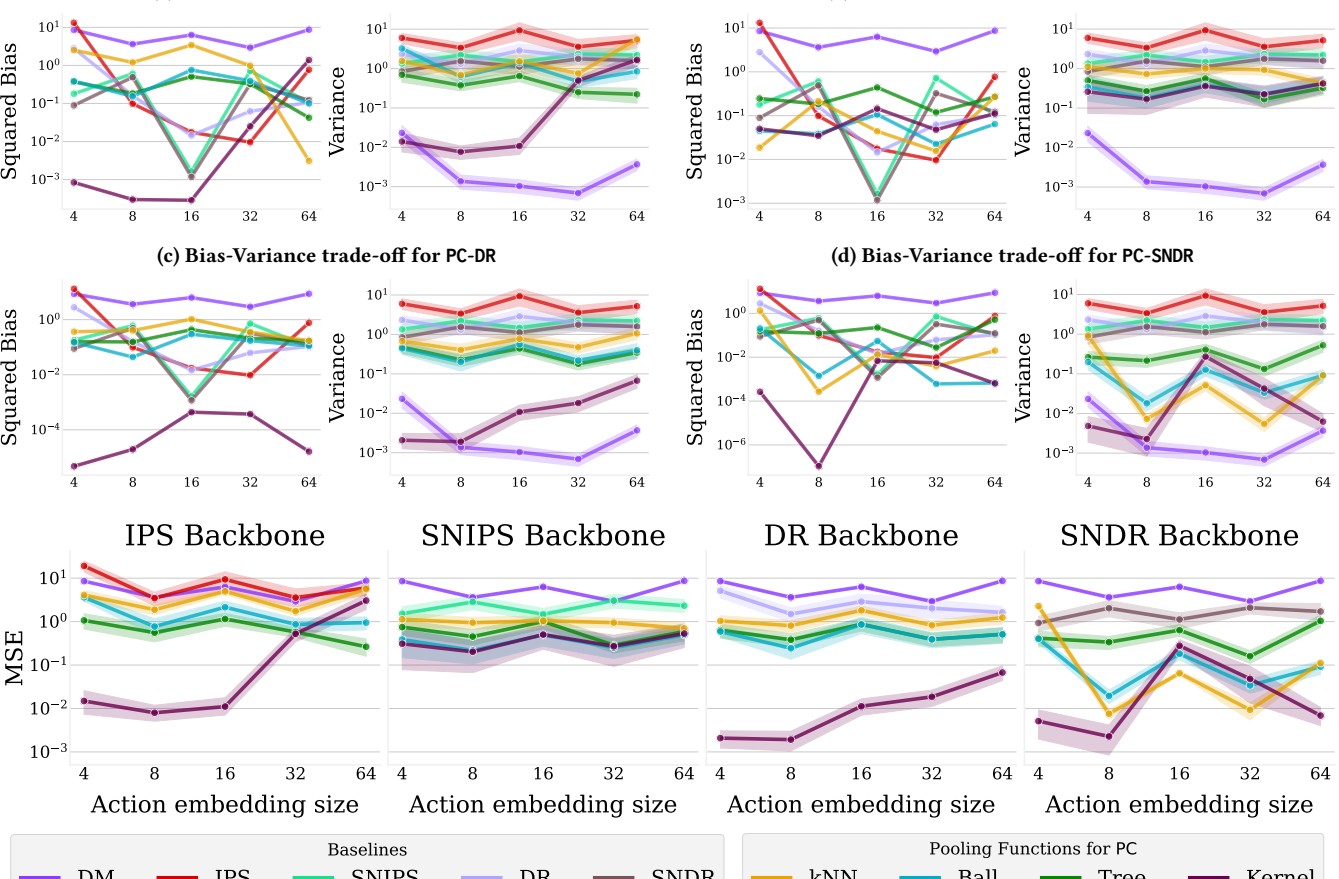

**Figure 23: Change in MSE, Squared Bias & Variance while estimating** $V(\pi_{\mathrm{good}})$ **with varying action embedding size (**log-log **scale) for the synthetic dataset (**2000 **actions), using data logged by** $\mu_{\mathsf{uniform}}$**.**

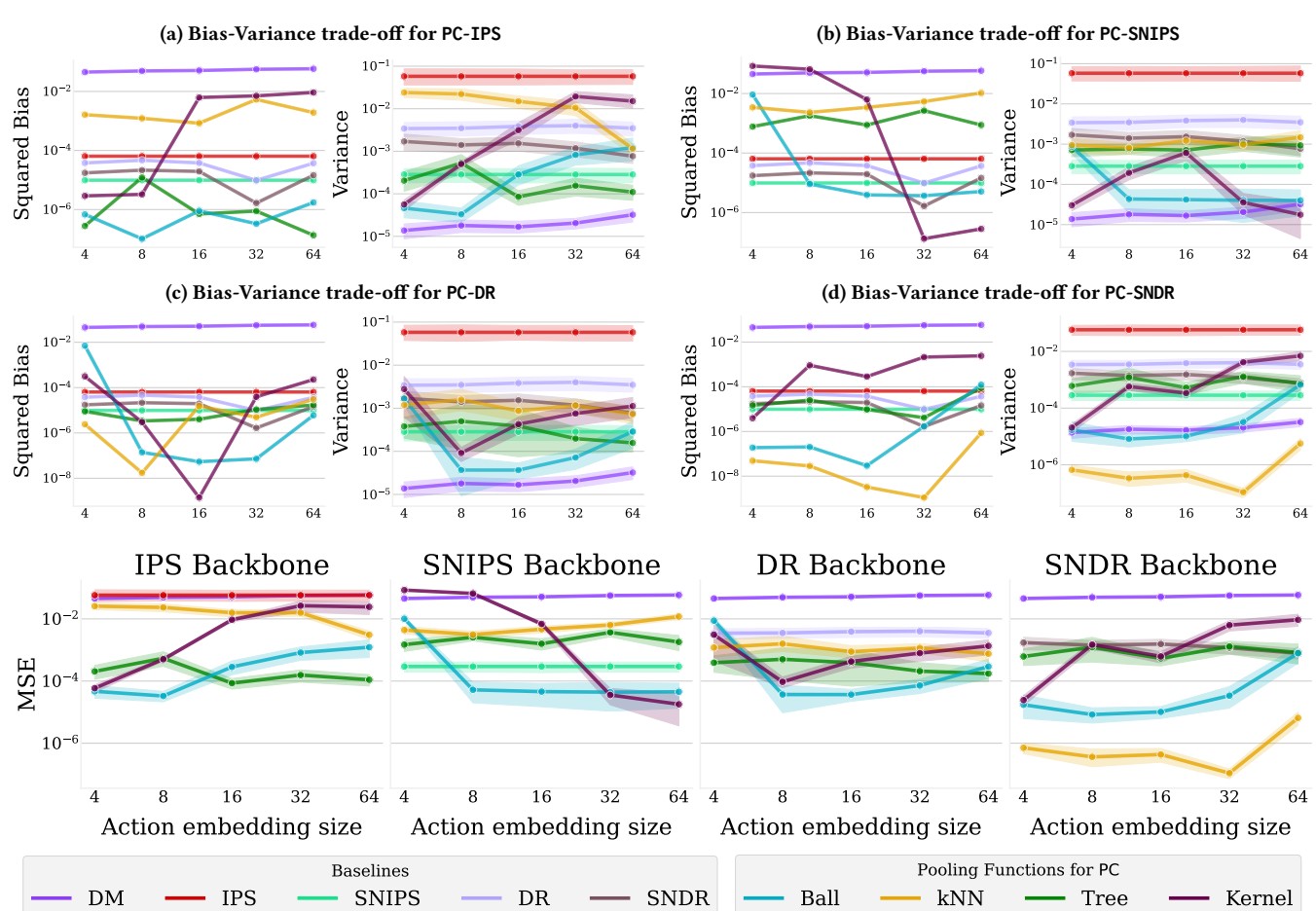

**Figure 24: Change in MSE, Squared Bias & Variance while estimating** $V(\pi_{\text{good}})$ **with varying action embedding size (**log-log **scale) for the movielens dataset, using data logged by** $\mu_{\text{uniform}}$**.**

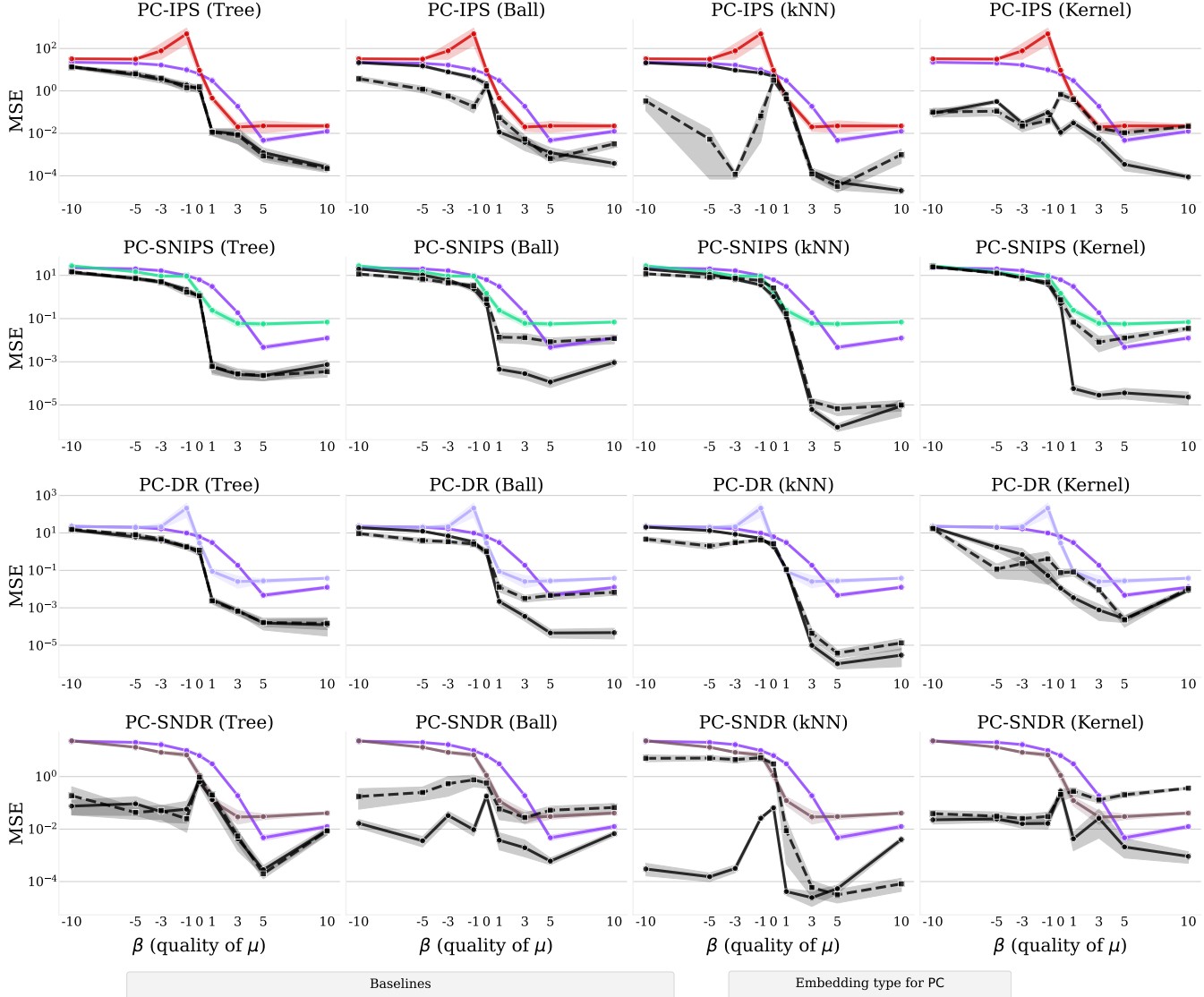

Figure 25: Change in MSE while estimating $V(\pi_{\text{good}})$ with varying logging policies (log-scale) and using PC with Oracle *vs.* Data-driven action embeddings for the synthetic dataset.

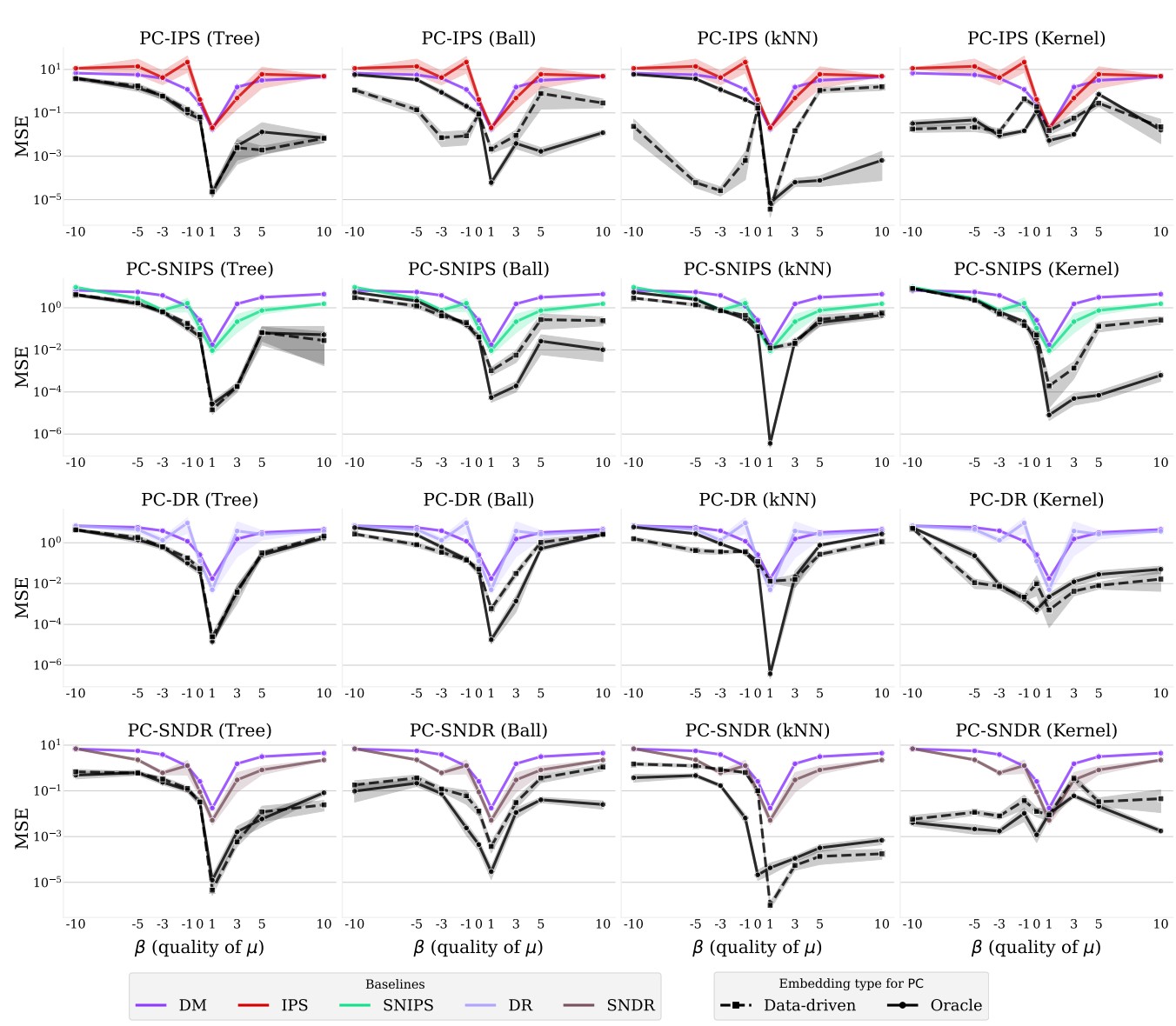

**Figure 26: Change in MSE while estimating $V(\pi_{\text{bad}})$ with varying logging policies (log-scale) and using PC with Oracle *vs.* Data-driven action embeddings for the synthetic dataset.**

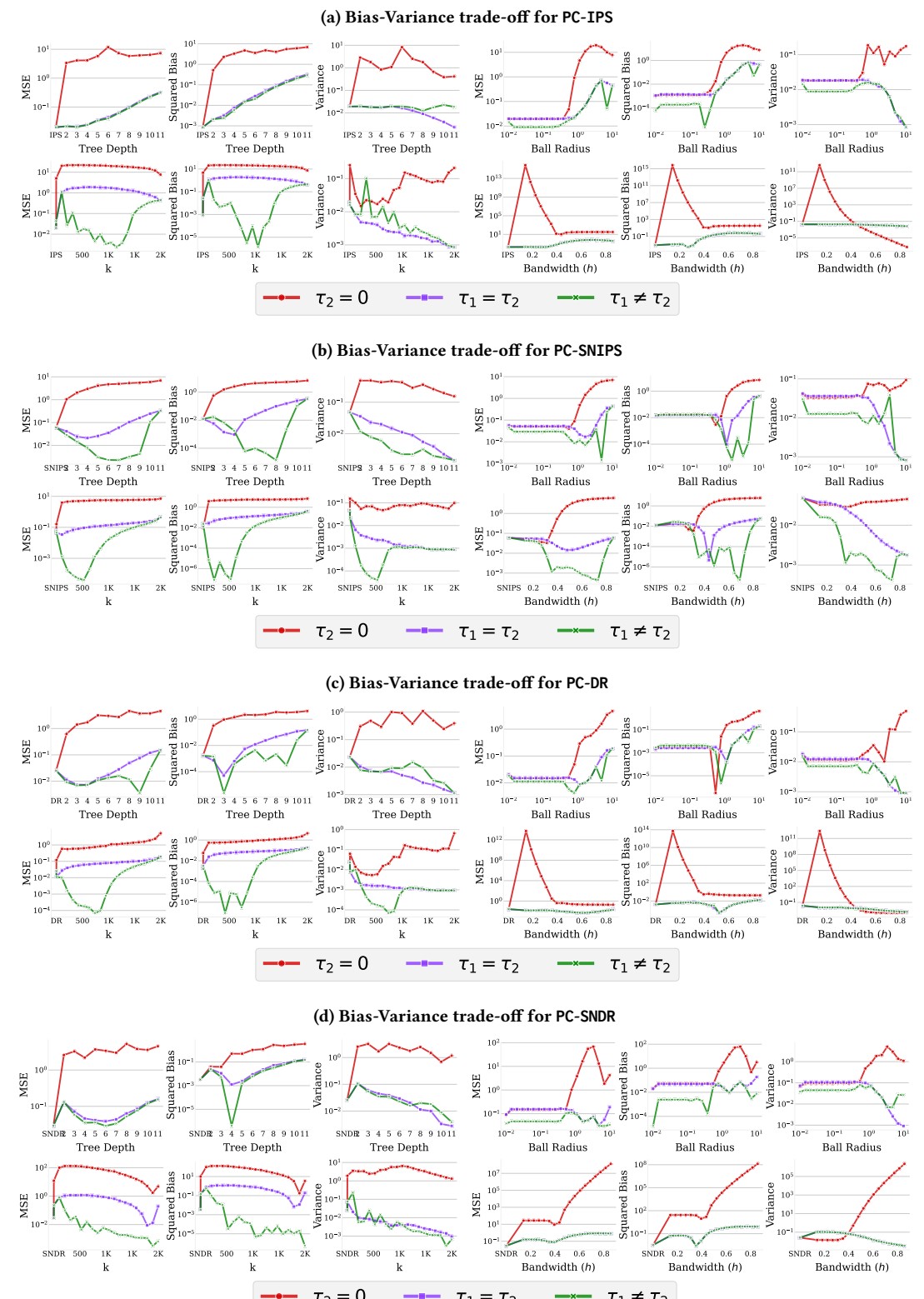

Figure 27: Visualizing the bias-variance trade-off for the PC estimator for different backbones & pooling strategies, while estimating $V(\pi_{\text{good}})$ with varying amount of pooling (log-log scale) on the synthetic dataset (with 2000 actions), using $\mu_{\text{good}}$ for logging. Note that the respective naïve backbone estimators are the left-most point in each plot, i.e., when there's no pooling.

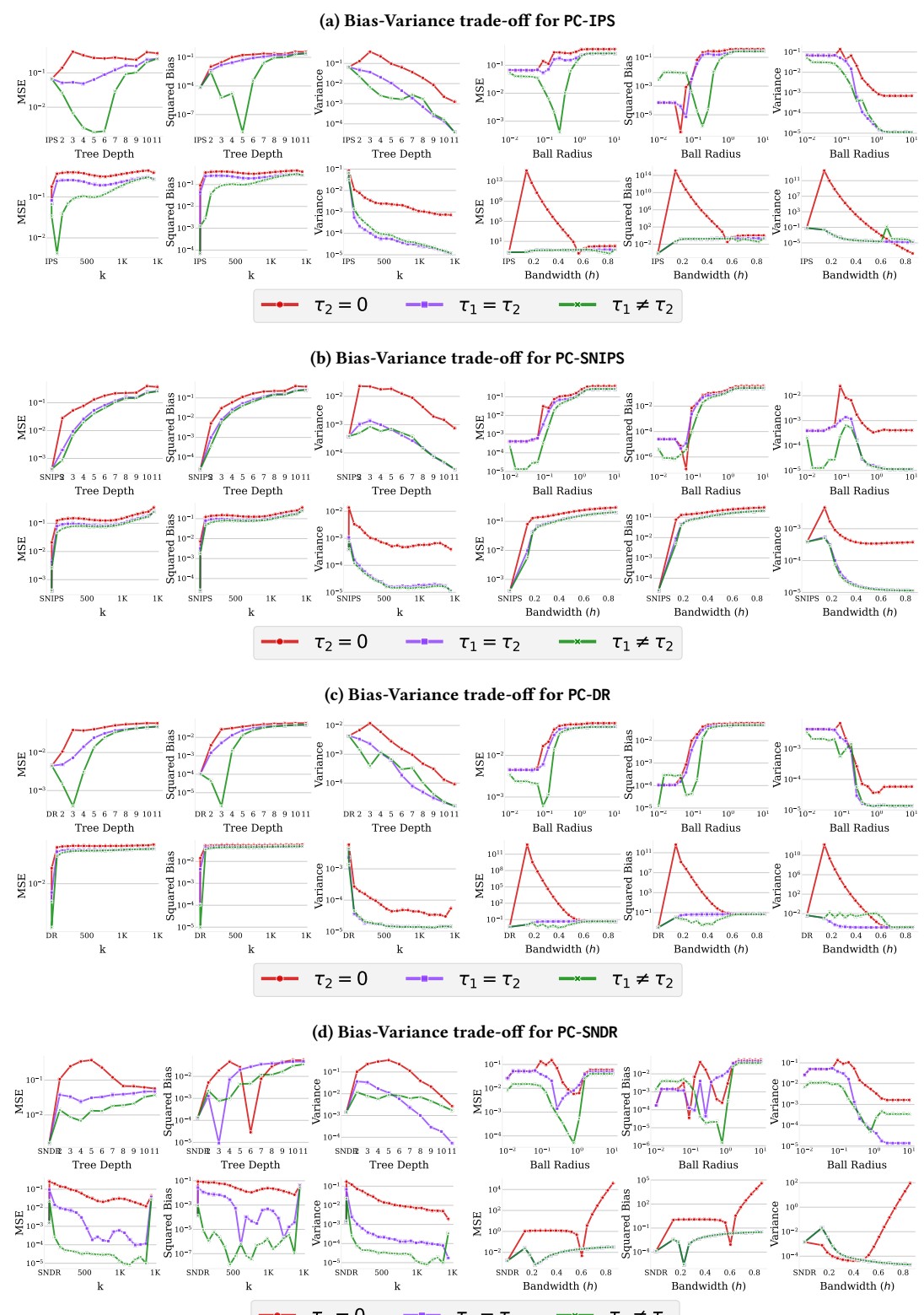

**Figure 28: Visualizing the bias-variance trade-off for the PC estimator for different backbones & pooling strategies, while estimating $V(\pi_{\text{good}})$ with varying amount of pooling ($\log$-$\log$ scale) on the movielens dataset, using $\mu_{\text{good}}$ for logging. Note that the respective naïve backbone estimators are the left-most point in each plot, *i.e.*, when there's no pooling.**

(a) Optimal amount of pooling with varying number of actions, *i.e.,* $|\mathcal{A}|$ (log **scale**)

(b) Optimal amount of pooling with varying amount of bandit feedback, *i.e.,* $|\mathcal{D}|$ (log **scale**)

Figure 29: Change in the optimal amount pooling for **PC** while estimating $V(\pi_{\text{good}})$ for the synthetic dataset, using data logged by $\mu_{\text{bad}}$.

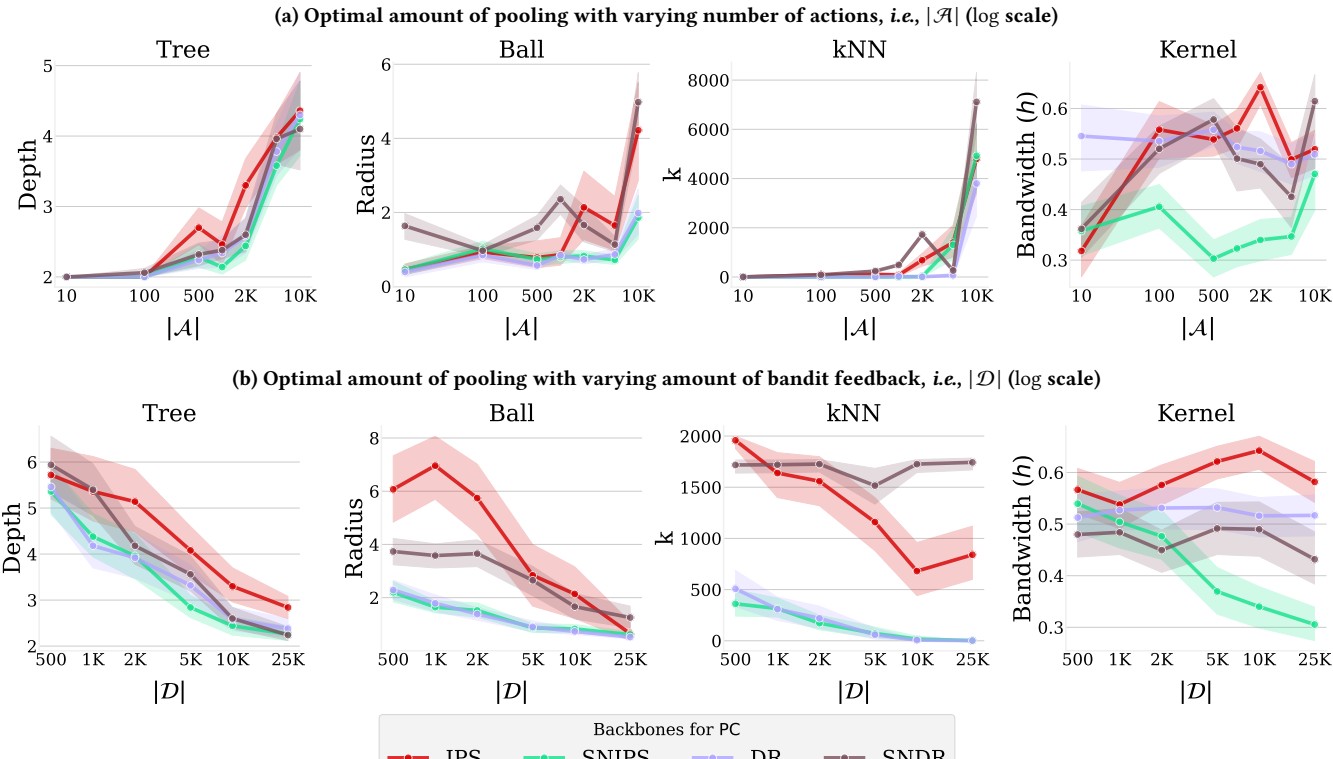

Figure 30: Change in the optimal amount pooling for PC while estimating $V(\pi_{\mathrm{good}})$ for the synthetic dataset, using data logged by $\mu_{\mathsf{uniform}}$.

**(a) Optimal amount of pooling with varying number of actions, *i.e.*, $|\mathcal{A}|$ (log scale)**

**(b) Optimal amount of pooling with varying amount of bandit feedback, *i.e.*, $|\mathcal{D}|$ (log scale)**

**Figure 31: Change in the optimal amount pooling for PC while estimating $V(\pi_{\text{good}})$ for the synthetic dataset, using data logged by $\mu_{\text{good}}$.**

