# OpenReview forum: "Off-Policy Evaluation for Large Action Spaces via Policy Convolution"
_ACM.org/TheWebConf/2024/Conference — TheWebConf24_

### Official Review · Reviewer_JoEa · 2023-11-02

**Novelty:** 6
**Technical Quality:** 6

**Review:**

The authors provide a comprehensive review of inverse propensity score estimators for off policy evaluation and propose a new estimator based on convolution that has greatly reduced variance as demonstrated in simulations and experiments in the realistic situation where the items have embeddings that can be used in order to identify similar actions.

In general I find the presentation to be very well done, the idea novel and the experimental valuation convincing.

A great many papers proposing IPS estimation improvements are based on the observations that the ratio of pi/mu can be very high and propose modifications to this ratio.  I think this is a rather limited view of the problem, and I am pleased that the authors of this paper introduce a different idea – that some actions are similar and we might reasonably expect the reward of these actions to be similar.  This adds bias to the estimation process, but greatly reduces variance and I find the arguments that it can be very beneficial compelling.

The core of this idea is to replace the ratio pi(a|x)/mu(a|x) with a new ratio pi’(a|x)/mu’(a|x) where pi’ and mu’ are new policies created by convolving the old policies.  The convolution is done with the benefit of additional information in the form of action embeddings.  Imagine a1 and a2 are similar actions, the logging policy puts most of the mass on a1, and almost no mass on a2.  The new policy we would like to evaluate puts almost no mass on a1 and almost no mass on a2.  With traditional IPS this will result in very high importance weights as 1/mu(a2|x) is very high.  The policy convolution estimator instead will perform a convolution – this means that brings both pi and mu closer together and therefore reduces variance.  There are differences in each of the two convolutions however.  A convolution on pi has the interpretation – instead of evaluating pi that is really different from the logging policy let’s evaluate a similar policy pi’ that is similar to it.  In contrast, a convolution on mu has the interpretation – as a1 and a2 are similar we can use reward estimated on a1 to learn about a2 and vice a versa.  It would seem to me that convolving or modifying mu would only be beneficial in cases where it made mu closer to pi.
In the end of the paper the authors compare the method with the marginal IPS estimator (Saito and Joachims 2022).  They make the correct observation that the sampling assumptions for the policy underlying this estimator are not realistic in practice.  Would the author’s agree that convoluted policy estimation is very similar to marginal IPS estimation – but recommends the use of the method in much broader circumstances?

A paper that the authors may have overlooked is Sakhi et al (2020).  Like the convolved estimator it also uses action-action similarities, but it also employs context-context similarities.  This is done in a direct method formulation via a prior distribution allowing the same treatment for both action-action and context-context similarities.  It would seem that an IPS based formulation must develop different methodologies for treating action-action similarities and context-context similarities as the estimator involves ratios pi(a|x)/mu(a|x) where the action and the context have different roles – in contrast direct methods regress the reward off the action and the context.  I don’t think it is necessary to do further experiments, but it seems plausible that Sakhi et al (2020)’s method may be competitive with convolved IPS.  This is because in addition to using the action-action distances it can also use context-context distances.  Furthermore, it does not violate the conditionality principle (Berger and Wolpert 1988).

As a final remark, I particularly like the way the authors are careful to highlight the assumption of un-confoundedness.  This assumption will typically be satisfied in production environments if all sub-models involved in decision making have access to the complete contextual features – but in practice this is often not the case.

**Questions:**

On line 445 the covariance denoted is given a normal distribution - this looks like an error (covariances are usually matrices/scalars variance must be positive).
There are a few issues I would be interested on the authors commentary on.

•	The whole formulation of off policy estimation concerns the reward averaged over the contexts – but in order to find the optimal policy this is neither necessary or sufficient – instead we need only the reward estimator as produced by the direct method.  Why then should we focus on correct estimation of “V”?  Should users of the direct method be concerned by the results of Robins & Ritrov (1997)?

•	The authors demonstrate their method with simulated data and experiments based on MovieLens.  MovieLens is not really a bandit problem, and it is a bit artificial to try to turn it into one.  What is the value of a MovieLens based experiment – except that some referees may like/demand them?

•	What are the differences between convolving pi and mu?  Should they be done in the same way?

•	How can context-context distances be brought into the proposed method?  Any commentary on how the method may compare with Sakhi et al (2020)?

•	Any commentary on the discussion above.

•	The size of the action space investigated here are still far below what industrial scale recommendation systems deal with.  Is it feasible to apply these methods when action spaces are in the millions or even higher?

**Reviewer Confidence:**

4: The reviewer is certain that the evaluation is correct and very familiar with the relevant literature

**Scope:**

4: The work is relevant to the Web and to the track, and is of broad interest to the community

---

### Official Review · Reviewer_EkM3 · 2023-11-21

**Novelty:** 4
**Technical Quality:** 4

**Review:**

This paper tries to address the challenge of off-policy evaluation, crucial for assessing and optimizing new policies. The primary issue tackled is the distribution shift between the logging policy (data generation) and the target policy (evaluation). The common approach of importance sampling, while providing unbiased estimates, often introduces high variance, especially in cases of large action spaces. To overcome this, the authors propose the Policy Convolution (PC) estimator, which strategically convolves logging and target policies using action embeddings, introducing a bias-variance trade-off that can be controlled. However, there are several questions in this paper, and see Questions below for details.

**Questions:**

1. This article tends to focus more on pure reinforcement learning and is not strongly connected to the "User Modeling and Recommendation" track. Although the introduction briefly mentions the work and its relation to the web and recommendation systems (only in one sentence), I feel that the content of the entire article rarely delves into the aspects of recommendation systems. At least, that's the impression I get from the current writing style of the article. Therefore, I am inclined to suggest that this paper might find a better fit in other conferences, such as ICML, NeurIPS, AAAI, etc., rather than WWW.

2. The datasets used in this article lack persuasiveness, including the use of synthetic data and a dataset that is not very recent. If the authors aim to demonstrate the strong performance of the proposed work, it would be advisable to conduct experiments on more up-to-date real-world data, preferably directly related to recommendation systems.

3. It's currently unclear how the proposed policy convolution is directly related to performance improvement. At least, the analysis provided so far doesn't seem comprehensive enough.

**Reviewer Confidence:**

3: The reviewer is confident but not certain that the evaluation is correct

**Scope:**

2: The connection to the Web is incidental, e.g., use of Web data or API

---

### Official Review · Reviewer_jX3e · 2023-11-30

**Novelty:** 4
**Technical Quality:** 4

**Review:**

Off-policy estimation (OPE) is a widely adopted approach in reinforcement learning that encounters the challenging of distribution shift between the logging policy, responsible for generating the data, and the target policy, particularly in scenarios with a large action space. In order to tackle this challenge, the present study introduces a novel Policy Convolution (PC) estimator that leverages latent action structure defined by action embeddings to enhance off-policy evaluation.

Pros:
- The draft is well-organized, providing concise and clear explanations of related works and OPE estimators.
- The experimental design is elaborate, with clearly defined objectives. The titles in Sec 4.2, formulated in the form of a problem, are easily understandable.

Cons:
- The presentation of the proposed method in this paper is quite brief, with Section 3 comprising only about one page. It is recommended to consider elaborating on the theoretical analysis of the method's "bias-variance trade-off" and providing a more comprehensive description of the motivating example.
- The experimental line chart in this paper contains multiple lines, and the lines are visually indistinguishable, making it difficult to differentiate between the proposed method and the baselines. It is recommended to consider using dashed lines for the baselines to differentiate them from the solid lines representing the proposed method.

**Questions:**

1. The experiments in this paper were conducted on only two datasets, with one of them being a self-constructed dataset. It is desirable to conduct experiments on a wider range of datasets to demonstrate the applicability of the proposed method.
2. The performance of the four pooling functions varies in different scenarios (i.e., different datasets and backbones). Some choices may appear to perform worse than the baseline. In practical usage of the PC estimator, how should one choose the pooling function?
3. How does the proposed method perform in terms of computational efficiency? Does it incur a significant time cost compared to the baseline?

**Reviewer Confidence:**

3: The reviewer is confident but not certain that the evaluation is correct

**Scope:**

3: The work is somewhat relevant to the Web and to the track, and is of narrow interest to a sub-community

---

### Official Review · Reviewer_MJi1 · 2023-12-01

**Novelty:** 6
**Technical Quality:** 5

**Review:**

__Summary__

This paper presents a new approach to the problem of off-policy evaluation (OPE) in the contextual bandit setting. Traditional methods for OPE, such as Importance Sampling, struggle with high variance when dealing with large action spaces or significant policy mismatches. The authors propose the Policy Convolution (PC) estimator to address these challenges. PC uses action embeddings to leverage latent structures within actions, hypothesizing that similar embeddings correlate with similar expected rewards. The goal of PC is to create a more favorable bias-variance trade-off by convolving the logging and target policies. The paper tests the PC estimator on both synthetic and real-world datasets, focusing on its performance in reducing mean squared error under conditions of large action spaces or high policy mismatch.

__Strengths__
- The concept of the Policy Convolution (PC) estimator is novel and addresses a well-defined gap in off-policy estimation tasks. The motivation behind the approach is clearly articulated, demonstrating a deep understanding of the challenges in this domain.
- The paper presents extensive ablative and controlled evaluations, showcasing the robustness of the PC approach in various off-policy estimation scenarios.
- The paper is well-structured and presents its ideas in an intuitive and straightforward manner, making it accessible and easy to comprehend. The logical flow and clarity of the presentation facilitate a quick grasp of the core concepts and methodologies.
- The assumptions underlying the PC approach are discussed well, providing clarity on the conditions and contexts in which the approach is most effective. This discussion helps in understanding the scope and limitations of the proposed method.

__Weaknesses__
- Presentation clarity: The figures in the paper are cluttered and difficult to interpret, with too many lines that obscure the underlying trends.
- Abstract specificity: The abstract does not clearly state that the paper's focus is exclusively on the contextual bandit problem.
- Theoretical justification: The claim that "offCEM" and "groupIPS" are special cases of PC lacks sufficient theoretical substantiation or derivation, at least in the appendix.
- Baseline comparisons: The evaluations primarily compare PC variants to basic policy estimators without considering baselines that also use latent action information, which is a key aspect of PC.

__A bit more detailed comments on the cons__

Firstly, the figures presented are somewhat cluttered and challenging to interpret, with an excess of overlapping lines that make it difficult to discern specific trends. A more streamlined approach to data visualization could greatly enhance the clarity of these results.

Additionally, the paper's claim on page 4 that "offCEM" and "groupIPS" are special cases of the proposed PC method lacks comprehensive theoretical substantiation. Including detailed derivations or expanded explanations, possibly in the appendix, would strengthen this assertion.

Lastly, while the evaluation compares PC variants to standard policy estimators, it overlooks the comparison with other methods that also use latent action information. Such comparisons, particularly with "offCEM" and "groupIPS," would provide a more robust validation of PC's effectiveness in contexts where latent action information is utilized.

**Questions:**

- Could you provide a formal derivation to clarify the relationships between PC and other methods, i.e., offCEM and groupIPS? This would help understand the theoretical underpinnings and the distinctions or similarities between these approaches.
- In Equation(1), the term $q(x, \cdot)$ is used for the logging policy $\mu(a|x)$. However, this term isn't defined elsewhere in the paper. Is this a typo of $\delta(x, \cdot)$?
- In Figure 2, several lines indicate a decrease in MSE as the action space size increases. This doesn't seem straightforward; could you explain why and how this trend occurs?
- Regarding the "DR Backbone" in Figure 2, there's a noticeable peak in MSE for the kNN PC variant. This peak represents a significant increase in MSE. What could be the underlying reason for this anomaly?
- On page 6, the discussion about Figure 4 mentions that "as $|\mathcal{D}|$ continues to increase, PC converges to its respective backbone estimator." However, this statement isn't clearly reflected in the figure. Could you provide further explanation or clarification on how this conclusion was drawn?
- The paper assumes that actions with similar rewards will be closer in the latent space. How would the efficacy of the PC be affected if this assumption does not hold? Discussing the impact of embedding quality on PC's performance would be valuable in understanding its robustness under varying conditions of the latent space.

**Reviewer Confidence:**

4: The reviewer is certain that the evaluation is correct and very familiar with the relevant literature

**Scope:**

4: The work is relevant to the Web and to the track, and is of broad interest to the community

---

### Decision · Program_Chairs · 2024-01-22

**Decision:**

Accept

**Comment:**

The paper proposes a novel approach to off-policy estimation by reducing variance through policy convolution. Reviewers find the idea highly novel and interesting. The author response is well-argued and thorough.

 One reviewer is concerned about relevance to the UMAP track. While I do not share this particular concern, I do agree with the reviewer's sentiment that the connection to recommendation systems can be easily missed if not reading the paper carefully. To this point, I find the discussion of the MovieLens experimental setup from lines 489-508 to be rather opaque and technical without carefully discussing the motivation for this setup or justifying why this experiment in particular is *the* experiment the authors should be running with actual recommendation data. I remark that *three* reviewers question the rationale for using MovieLens (only) and how the setup relates to recommendation. The authors provide useful justification in their rebuttal that shuold be incorporated into the experimental design description along with more explanatory detail (not just what was done, but why it was done).

 In addition, a large number of other points of clarification and minor correction came up in the reviews and I think it is critical for the authors to incorporate their responses and suggested changes on revision. All of these revisions seem straightforward and only relate to presentation.

 With revisions as suggested above (and in the reviews), I believe this paper can make a solid and novel contribution to the WWW UMAP track.